

# Characterizing the marine iodine cycle and its relationship to ocean deoxygenation in an Earth System model

Keyi Cheng[1], Andy Ridgwell[2], Dalton S. Hardisty[1]

[1]Department of Earth and Environmental Sciences, Michigan State University, East Lansing, 48823, USA

[2]Department of Earth and Planetary Sciences, University of California Riverside, Riverside, 92521, USA

*Correspondence* to: Keyi Cheng (chengkey@msu.edu)

**Abstract.** Iodine abundance in marine carbonates (as an elemental ratio with calcium – I:Ca) is of broad interest as a proxy for local/regional ocean redox. This connection arises because the speciation of iodine in seawater—in terms of the balance between iodate ($IO_3^-$) and iodide ($I^-$)—is sensitive to the prevalence of oxic vs. anoxic conditions. However, although I:Ca ratios are being increasingly commonly measured in ancient carbonate samples, a fully quantitative interpretation of this proxy is hindered by the scarcity of a mechanistic and quantitative framework for the marine iodine cycle and its sensitivity to the extent and intensity of ocean deoxygenation. Here we present and evaluate a representation of marine iodine cycling embedded in an Earth system model ('cGENIE') against both modern and paleo observations. In this, we account for $IO_3^-$ uptake and reduction by primary producers, the occurrence of ambient $IO_3^-$ reduction in the water column, plus the re-oxidation of $I^-$ to $IO_3^-$. We develop and test a variety of different mechanistic relationships between $IO_3^-$ and $I^-$ against an updated compilation of observed dissolved $IO_3^-$ and $I^-$ concentrations in the present-day ocean. In optimizing the parameters controlling previously proposed mechanisms behind marine iodine cycling, we find that we can obtain broad matches to observed iodine speciation gradients in zonal surface distribution, depth profiles, and oxygen deficient zones (ODZs). We also identify alternative, equally well performing mechanisms which assume a more explicit mechanistic link between iodine transformation and environment. This mechanistic ambiguity highlights the need for more process-based studies on modern marine iodine cycling. Finally, because our ultimate motivation is to further our ability to reconstruct ocean oxygenation in the geological past, we conducted 'plausibility tests' of our various different model schemes against available I:Ca measurements made on Cretaceous carbonates – a time of substantially depleted ocean oxygen availability compared to modern and hence a strong test of our model. Overall, the simultaneous broad match we can achieve between modelled iodine speciation and modern observations, and between forward-proxy modelled I:Ca and geological elemental ratios supports the application of our Earth system modelling in simulating the marine iodine cycle to help interpret and constrain the redox evolution of past oceans.

## 1. Introduction

Dissolved Iodine (I) in seawater is redox sensitive and as such, a potential invaluable delineator of past ocean deoxygenation. This arises directly from: (1) observations that the oxidized iodate ($IO_3^-$) is reduced to iodide ($I^-$) under low oxygen conditions, and (2) that $IO_3^-$ in seawater is incorporated into carbonate lattice during precipitation in proportion to its seawater abundance (whilst $I^-$ is not) (Lu et al., 2010; Podder et al., 2017; Kerisit et al., 2018; Zhang et al., 2013; Hashim et al., 2022). Hence, past ocean $IO_3^-$ concentrations can be recorded in coeval carbonates as I:Ca



ratios, with the potential of carbonate I:Ca to reflect the redox variation of the ancient seawater (Lu et al., 2010).
Indeed, the I:Ca ratio in marine carbonates is already widely applied as a paleoredox proxy, with studies employing it
to explore the $[O_2]$ variation throughout much of Earth history, from the Archean and through the Cenozoic (Lu et al.,
2010; Hardisty et al., 2014; Zhou et al., 2015; Lu et al., 2016; Edwards et al., 2018; Lu et al., 2018; Bowman et al.,
2020; Pohl et al., 2021; Wei et al., 2021; Ding et al., 2022; Shang et al., 2019; Liu et al., 2020; Fang et al., 2022;
Uahengo et al., 2020; Yu et al., 2022; Tang et al., 2023). The potential for I:Ca to generate critical insights into how
the oxygenation of the ocean has evolve through time, as well as the causes and biological/ecological consequences
of this, requires that we have an adequate understanding, not only of carbonate $IO_3^-$ incorporation, but of the marine
iodine cycle in general.

Progress has been made towards understanding the marine iodine cycle in the past decades. Iodine has a long

residence time (~300 kyr; Broecker and Peng, 1983), making its concentration among the global ocean relatively
constant around 500nM (Elderfield and Truesdale, 1980; Truesdale et al., 2000; Chance et al., 2014). Although the
total concentration is relatively invariant, the two most abundant species of dissolved iodine in the ocean, $IO_3^-$ and
iodide $I^-$, vary relative to each other depending on the environment. Today, $IO_3^-$ is generally the dominant iodine
species in oxygenated regions of the ocean, representing total iodine nearly quantitatively below the euphotic zone.
Within the euphotic zone, $I^-$ occurs and generally increases in association with release during phytoplankton growth
and senescence (Hepach et al., 2020). Iodide is also more abundant in oxygen deficient zones (ODZ) – often but not
always quantitatively so (Truesdale et al., 2000; Rue et al., 1997; Cutter et al., 2018; Moriyasu et al., 2020; Farrenkopf
and Luther, 2002; Wong and Brewer, 1977; Truesdale et al., 2013; Rapp et al., 2020, 2019). Respectively, within the
ODZ, $IO_3^-$ is reduced to $I^-$ and hence has low concentrations (Rue et al., 1997; Farrenkopf et al., 1997; Moriyasu et al.,
2020; Rapp et al., 2019, 2020).

Although generally depleted in low-$[O_2]$ settings, the causal spatial relationship between seawater $[O_2]$ and

$[IO_3^-]$ is not simple and is currently not well understood. Recently published observations from global oxygen deficient
zones (ODZ) reveals that the relationship between dissolved $[O_2]$ and $[IO_3^-]$ is not linear, but instead it is possible that
there is a certain $[O_2]$ or related redox threshold triggering $IO_3^-$ reduction (Cutter et al., 2018; Moriyasu et al., 2020;
Farrenkopf and Luther, 2002; Rue et al., 1997; Chapman, 1983). Dissimilatory $IO_3^-$ reducing bacteria, as well as
abiotic reduction with sulfide and dissolved Fe, have been identified within ODZs (Farrenkopf et al., 1997; Councell
et al., 1997; Jiang et al., 2023). In addition, slow oxidation-reduction kinetics (Tsunogai, 1971; Hardisty et al., 2020;
Schnur et al., 2024) imply the likelihood that *in situ* iodine signals could be integrated across large-scale physical
oceanography processes – including ocean currents and mixing between water masses (Hardisty et al., 2021), and
meaning that iodine species reflect regional rather than local redox conditions (Lu et al., 2020b). Non-redox related
processes, such as phytoplankton-mediated $IO_3^-$ reduction and organic matter remineralization also exerts controls on
iodine speciation in the water column (Fig. 1; Elderfield and Truesdale, 1980; Wong et al., 1985; Luther and Campbell,
1991; Hepach et al., 2020). Therefore, it is difficult to infer water column redox simply based on iodine speciation
without considering these physical and biological effects.



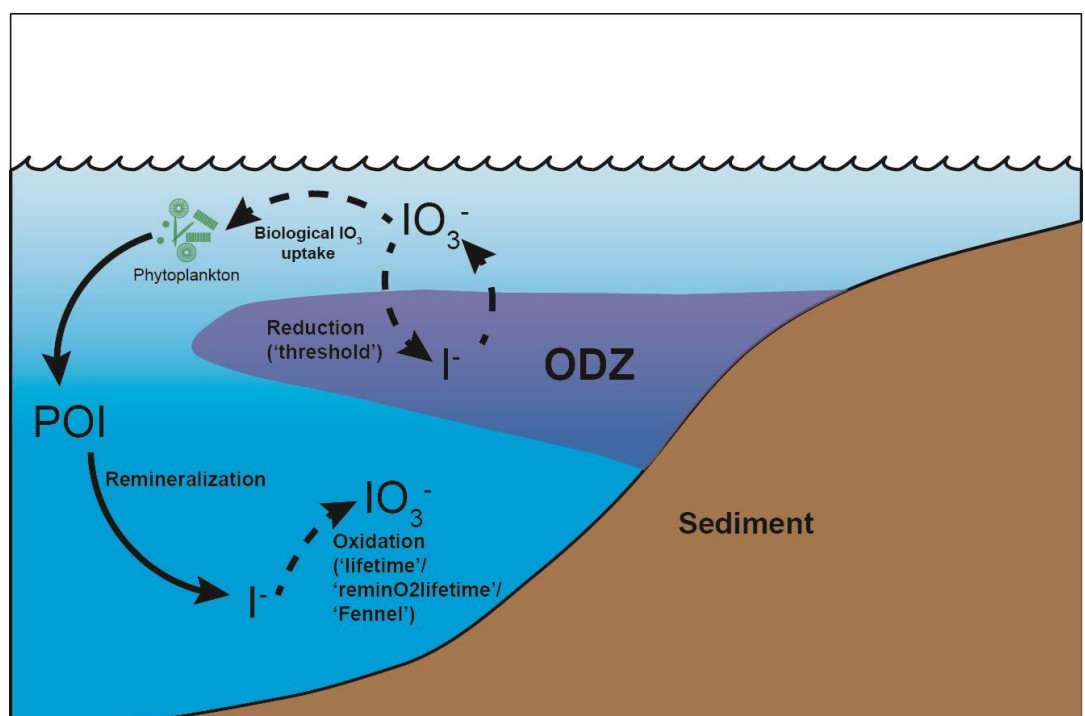

**Figure 1: The iodine cycle in marine oxygen deficient zones (ODZ) in cGENIE. The oxidation-reduction options ('threshold' and 'lifetime'/ 'reminO2lifetime'/ 'Fennel') are described in Section 2.2 and Table 1. Dashed arrows indicate variable processes during ensemble simulations. Note that the POI export is controlled by temperature (TDEP).**

Apart from the uncertainties associates with $IO_3^-$ reduction, it is notable that the oxidants responsible for $IO_3^-$ formation during the re-oxidation $I^-$ are currently unknown, only that it is unlikely to be $O_2$, which is not thermodynamically favorable to oxidize iodine (Luther et al., 1995). A recent thermodynamic review indicates that the reactive oxygen species (ROS) such as hydrogen peroxide and OH radicals can fully oxidize $I^-$ to $IO_3^-$. Iodide oxidation to $IO_3^-$ is a 6-electron transfer and other ROS, such as superoxide, are only thermodynamically favorable to catalyze partial oxidation to intermediates (Luther, 2023). These ROS species have heterogenous distributions and ambient ocean concentrations that are typically relatively low compared to iodine, supporting the likelihood of temporally or spatially isolated high $I^-$ oxidation rates despite of overall extremely slow rates (Schnur et al., 2024). Additional support for spatially or temporally heterogenous $I^-$ oxidation rates comes from recent experimental observations of $IO_3^-$ production from $I^-$ in nitrifying cultures (Hughes et al., 2021). Nitrification rates vary globally, with the highest values occurring in ODZs and the dissolved chlorophyl maximum (summarized in Table 2 of Hughes et al., 2021). Regardless, nitrification or other specific mechanisms have yet to be linked directly to $I^-$ oxidation under normal marine conditions, leaving open the question of rates and locations of $I^-$ oxidation.

Given the prevailing uncertainty in the mechanisms governing the marine iodine cycle mentioned above, and in conjunction with I:Ca being a relatively new proxy, it is perhaps not surprising that few attempts have been made





to model the marine iodine cycle. In a recent publication, a model was developed to simulate modern ocean surface I⁻
distributions, with the aim of being able to improve tropospheric ozone models (Wadley et al., 2020). This particular
model was based around a relatively high horizontal ocean resolution (1° grid size) with a 3-layer vertical water
column. Iodine biogeochemical cycling was coupled with the nitrogen cycle, with the surface I⁻ distribution sensitive
to biological and hydrological factors including primary productivity, I:C ratio, oxidation, mixed layer depth,
advection, and freshwater flux. Because the Wadley et al., (2020) model was specifically focused on near-surface
processes within the upper 500 m, it did not consider processes occurring within ODZs and hence is not directly
applicable to questions concerning the controls on I:Ca ratios. In contrast, a second model-based study deliberately
targeted paleoceanographic questions and incorporated an iodine cycle including redox-controlled biogeochemical
reactions into the 'cGENIE' Earth system model (Lu et al., 2018). The advantage for paleo studies afforded by this
particular approach is that the cGENIE model can take into account different continental configurations, non-modern
atmospheric composition ($pO_2$, $pCO_2$), and other boundary conditions that may have differed on ancient Earth relative
to today (Ridgwell et al., 2007; Reinhard et al., 2016; Boscolo-Galazzo et al., 2021; Remmelzwaal et al., 2019; Pohl
et al., 2022; Reinhard and Planavsky, 2022).

Despite a growing understanding of I:Ca variations through geologic time, it remains challenging to

determine mechanisms driving spatiotemporal marine $[IO_3^-]$ and the degree that these are linked to seawater oxygen.
Hence, the proxy is qualitative or semi-quantitative. Here, we calibrate the iodine cycle within the cGENIE Earth
System model to provide a mechanistic framework for interpreting ancient I:Ca variations. In this study, we build on
the work of Lu et al., 2018 and further develop and test a series of new potential parameterizations for water column
iodine oxidation, reduction, cellular uptake, and release during remineralization. We also developed 3 criteria for
assessing the model: (1) Statistical evaluation using the 'model skill score' (Watterson, 1996)— a non-dimensional
measure calculated using location-dependent comparisons between the model and an iodine ocean observation data
compilation. (2) Graphical comparison of modeled and observed iodine across 3 illustrative iodine speciation gradients
(depth profiles from multiple ocean basins, latitudinal transects of surface waters, and across transects of the Eastern
Tropical North Pacific oxygen minimum zone (Moriyasu et al., 2020)). (3) Model applicability to ancient settings by
comparing projections of ocean surface I:Ca with published I:Ca data from the Cretaceous (Zhou et al., 2015).
**2.        Model Description**
**2.1      The cGENIE Earth system modelling framework**
cGENIE is a class of model known as an 'Earth system model with intermediate complexity' (EMIC)—a global
climate-carbon cycle model that simplifies one or more (typically physical climatic) components of the Earth system.
In the case of cGENIE, ocean circulation is solved for on a relatively low-resolution grid: an equal area 36×36 grid,
which equates to 10° in longitude and latitude increments from 3° near the equator to 20° near the poles, and with 16
non-equally spaced vertical levels. This is coupled to a 2D energy-moisture-balance-model (EMBM) and a 2D
dynamic-thermodynamic sea-ice model. The physics are described in (Marsh et al., 2011; Edwards and Marsh, 2005).
We use a parameter calibration of seasonal pre-industrial climate following Cao et al., (2009).



The primary factors controlling the oceanic iodine cycle—specifically, biological productivity,
remineralization, and water column redox—are all represented in the model and described in Ridgwell et al., (2007).
In that particular configuration, the rate of organic matter export from the ocean surface is calculated based on just a
single nutrient (phosphate) control (together with light and sea-ice cover) and assumes a Redfield-ratio stoichiometry
with carbon (Fig. 1). Organic matter is split into particulate (POM) (33% of total export) and dissolved form (DOM)
(67%), with the former sinking down through the water column where it is progressively remineralized according to
a prescribed fixed 'decay' curve, while the latter is physically transported by circulation and decays (is remineralized
with a lifetime of 0.5 years). Here, we deviate from Ridgwell et al., (2007) (as well as the calibrated seasonal
configuration of Cao et al., 2009) by adopting a calibrated temperature-dependence to both export production as well
as the decay of POM in the water column (described in Crichton et al., 2021 and Boscolo-Galazzo et al., 2021).
Here, we use sulphate ($SO_4^{2-}$) as an electron acceptor supporting the remineralization of organic matter (both
POM and DOM) is governed by a $SO_4^{2-}$ half-saturaion limitation term as well as dissolved oxygen ($O_2$) inhibition,
while the rate of oxic resperation of organic matter is restricted by an [$O_2$] half-saturaion limitation term (as described
in Reinhard et al., 2020). This deviates from the framework described in Ridgwell et al., (2007). The difference is that
here, $SO_4^{2-}$ can be consumed even before dissolved oxygen can become fully depleted. Ambient temperature dictates
the total fraction of POM that decays through both pathways per unit time and within each ocean depth layer (Crichton
et al., 2021), with local [$O_2$] and [$SO_4^{2-}$] determining the fractional split between alternative pathways (Reinhard et al.,
2020). For DOM, the decay constant determines the total fraction that decays per unit time. It should be noted that
currently, there is no published nitrogen cycle in the cGENIE framework and hence we do not consider nitrate
reduction as part of the redox cascade.

## 2.2    Marine iodine cycling in cGENIE

In the cGENIE model, iodine is present in three reservoirs: $IO_3^-$ and $I^-$ in the water column, and $I^-$ incorporated in POM
(and DOM). We then consider four processes that transfer iodine between these reservoirs: (1) $IO_3^-$ reduction in the
water column, $I^-$ oxidation (also in the water column), photosynthetic $IO_3^-$ uptake and intercellular reduction to $I^-$, and
$I^-$ release to seawater during the remineralization of POM (and DOM) (Fig. 1). As dissolved tracers, $IO_3^-$ and $I^-$ are
physically transported and mixed through ocean circulation (as is I incorporated into dissolved organic matter),
whereas iodine in POM settles through the water column. This is effectively the same framework as used by Lu et al.,
(2018). Here we re-assess this framework against an updated compilation of observed iodine speciation in the modern
ocean and develop and test alternative representations of $IO_3^-$ reduction ("threshold", "inhibition" and
"reminSO4lifetime") and $I^-$ re-oxidation ("lifetime", "Fennel", and "reminO2lifetime").
In the numerical scheme of Lu et al., (2018), when [$O_2$] falls below a set concentration threshold, $IO_3^-$ is
immediately and quantitatively reduced to $I^-$ (thereafter, we term this iodate reduction parameterization "threshold").
The "inhibition" scheme links the $IO_3^-$ reduction rate with the ambient $O_2$ concentration. Following the
formulation for the rate of $SO_4^{2-}$ reduction in Reinhard et al., (2020), we apply an oxygen inhibition term governed by
a half-saturation constant. In devising this scheme, we note that while $IO_3^-$ reduction rates have been determined



experimentally, the quantitative relationship with $[O_2]$ (or other parameters) is unknown. The $IO_3^-$ reduction under
"inhibition" is mathematically described as:
$$d[IO_3^-]/dt = [IO_3^-] \times k_{red} \times \frac{k_{O_2}}{k_{O_2}+[O_2]} \quad (1)$$
in which $k_{red}$ is the maximum first-order reduction rate of $IO_3^-$, and $k_{O2}$ is the half-saturation constant of $O_2$.
Reduced sulfur (e.g. sulfides) is also suspected to play an important role in $IO_3^-$ reduction in seawater,
especially in the sulfidic zones (Jia-zhong and Whitfield, 1986; Luther and Campbell, 1991; Wong and Brewer, 1977;
Truesdale et al., 2013). We therefore devise a scheme ("reminSO4lifetime") that scales a nominal 'lifetime' for $IO_3^-$
with the rate of $SO_4^{2-}$ reduction in the model. This has the effect of increasing the rate of $IO_3^-$ reduction (a shorter
lifetime) under conditions of higher sulphate reduction rates and hence lower ambient oxygen concentrations and/or
higher rates of organic matter degradation:
$$d[IO_3^-]/dt = [IO_3^-] \times \frac{1}{\tau_{sul}} \times d[SO_4^{2-}]/dt \quad (2)$$
in which $\tau_{sul}$ defines the rate constant parameter linking the $IO_3^-$ and $SO_4^{2-}$ reduction, while the $d[SO_4]$ is amount of
$SO_4^{2-}$ reduced during each model timestep.
In Lu et al., (2018), $I^-$ is oxidized to $IO_3^-$ following first-order kinetics regardless of ambient $O_2$ (scheme
"lifetime"). In this scheme, $I^-$ oxidation follows the first-order reaction kinetics:
$$d[I^-]/dt = [I^-] \times \frac{1}{\tau} \quad (3)$$
where $\tau$ is the lifetime of $I^-$ in seawater.
Given the potential link between $I^-$ and nitrification, we devise an alternative "Fennel" scheme, in which $I^-$
oxidation rates vary as a function of ambient $O_2$, increasing with ambient $O_2$ concentrations towards some hypothetical
maximum value following Michaelis–Menten kinetics (Fennel et al., 2005). In Fennel et al., (2005), this
parameterization was originally devised for ammonia reoxidation. The form of this response is defined by the
maximum reaction rate and $O_2$ half-saturation constant (Fennel et al., 2005):
$$d[I^-]/dt = [I^-] \times k_{ox} \times \frac{[O_2]}{k_{fenn}+[O_2]} \quad (4)$$
in which $k_{ox}$ defines the maximum rate constant of $I^-$ oxidation, while $k_{fenn}$ is the $O_2$ half-saturation constant.
Finally, in "reminO2lifetime", we associate $I^-$ oxidation with $O_2$ consumption during remineralization. The
logic behind this parameterization is the recent observation of $I^-$ oxidation to $IO_3^-$ catalyzed by bacteria, perhaps in
association with ammonia oxidation (Hughes et al., 2021). Although the nitrogen cycle is not currently included in
cGENIE, the $NH_4^+$ oxidation can be scaled to OM remineralization (Martin et al., 2019) and hence to $O_2$ consumption
during remineralization. Under "reminO2lifetime", the lifetime of $I^-$ oxidation is inversely linked to $O_2$ consumption
so that faster remineralization—which in the ocean leads to more intensive $NH_4^+$ oxidation—enhances $I^-$ oxidation.
This $I^-$ oxidation scheme follow this equation:
$$d[I^-]/dt = [I^-] \times \frac{1}{\tau_{O2}} \times d[O_2]/dt \quad (5)$$
where $\tau_{O2}$ is the rate constant parameter and $d[O_2]$ is the $O_2$ consumption during remineralization during a single
timestep in the model.



The final process in the marine iodine cycle framework concerns the processing of iodine directly
through the biological pump. In this, $IO_3^-$ is taken up by phytoplankton and incorporated into OM during
photosynthesis (Elderfield and Truesdale, 1980) before being released as $I^-$ during remineralization and/or cell
senescence (Wong et al., 2002; Hepach et al., 2020; Wong et al., 1985). cGENIE simulates these processes as a
function of a 'Redfield-ratio' of iodine to carbon (I:C ratio) in OM. We note that while I:C is tunable, it is fixed
throughout the ocean. We discuss the merits of an optimized and uniform I:C compared to variable I:C (e.g., Waddley
et al., 2020) in more detail in the discussion.
**2.3    Model-data comparison**
We used the model skill measure (M) (Watterson, 1996) to assess the performance of the marine iodine cycle in
cGENIE. A major advantage of the M-score is that it is calculated through location-dependent comparison (Fig. 2; Lu
et al., 2018). Another advantage of the M score is that it captures overall improvement of model performance relative
to model-minus-observation maps, since it is non-dimensional, and the higher M stands for better performance. For
comparison with simulated distributions of iodine speciation, we compiled oceanic iodine observation data from the
literature (Fig. 2B; Table S4). This dataset includes the compiled dataset of Chance et al., (2019) and Sherwen et al.,
(2019), which was used to calibrate the Wadley et al., (2020) model, but includes more recent publications (referenced
in Table S4) and is expanded to include the deep ocean and ODZ data. To avoid the influence of freshwater dilution
and recycled iodine from the sedimentary flux, we applied a filter which only keeps the measurements with total iodine
between 450nM and 550nM in the dataset. Note that the $I^-$ measurements from the GP16 cruise in the ETSP are not
included for the comparison because of potential method considerations (see Cutter et al., 2018 and Moriyasu et al.,
2023). After these filtration methods, the data were re-gridded by taking the average values according to cGENIE
grids. For each iodine speciation (hereby $IO_3^-$ and $I^-$), a M score is calculated through comparing re-gridded
observations versus model results in each corresponding grid. The synthesized M score for iodine of each model
experiment is calculated through averaging those for both $I^-$ and $IO_3^-$.

**Figure 2: A). An example of location-dependent comparison between I- distributions in the cGENIE model iodine data array and the regridded ocean observation. B). The sampling locations of iodine observation data used for model-data comparison. Some coastal stations included in the figure are filtered out in the model-data comparison. The ETNP transect associated with Fig.6 is labeled as red box.**



**2.4      Sensitivity analyses and model implementation**
Because the relative roles of $IO_3^-$ reduction, $I^-$ oxidation, and $IO_3^-$ planktonic uptake in the water column are uncertain,
we calibrate the parameters controlling these processes in cGENIE by creating an ensemble of different parameter
value combinations arranged in a 2D regularly-spaced grid and then repeat the same 2D parameter ensemble for
different assumptions of I:C (Fig. 3). The output of each ensemble member is then statistically compared to our
observational database. We assume the associated parameterization when the model reaches the best M score of
replicating modern ocean iodine distribution would also be applied to simulate iodine cycling in the past. Each
ensemble member was run for a total of 2,000 years and each starts from the same initial state, which was an
experiment run for 10,000 years to equilibrium using a random set of iodine parameters within the ranges in Table 1.
Running the models for 2000 years minimizes the CPU time but was also found to be more than sufficient to allow
iodine inventories to equilibrate to new steady states. To explore whether the model simulated dissolved oxygen
distribution imparted any particular bias to the tuned iodine cycle, we repeated the model ensembles, continually
restoring the 3D pattern of $[O_2]$ in the model to that of the World Ocean Atlas 18 (WOA18) climatology (Garcia et
al., 2018).



**Figure 3: The three-dimensional model skill score array of the experiment ensembles.**

We ran model ensembles to test five different combinations of iodine cycling parameterizations – "lifetime-threshold",
"Fennel-threshold", and "reminO2lifetime-threshold", "lifetime-reminSO4lifetime", "lifetime-inhibition" (Table S1).
While the results for all parameterization combinations are given in the supplement, we only focus on 3
parameterization-combinations here (Table 1) — "lifetime-threshold", "Fennel-threshold", and "reminO2lifetime-



threshold". (A detailed justification and discussion for selecting these parameterization-combinations is included in
the Discussion section.)

**Table 1. The cGENIE iodine redox options and the associated range of parameters of these options.**
**The detailed introduction of each parameter is described in section 2.2.2 and the plausibility of these parameter**
**ranges is discussed in 4.1.1. Note that the oxidation rate constant $k$ in 'Fennel' is in unit of year$^{-1}$ in the model**
**configuration, which is the reciprocal of the 'lifetime'. A detailed table containing all considered**
**parameterization ranges can be found in Table S1.**

| Parameter description | | Iodine oxidation parameters | | | Iodine reduction parameters | I:C ratio ($\times 10^{-4}$ mol/mol) |
|---|---|---|---|---|---|---|
| | | 'lifetime' (years) | 'reminO2lifetime' ($\times 10^{-5}$ mol/kg) | 'Fennel' (Inhibition constant/ $\mu M\ O_2$) | 'threshold' ($\mu M\ O_2$) | |
| Simulation 1 | cGENIE $O_2$ | 10-170 | \ | \ | 1-110 | 0.5-3.5 |
| | WOA | 10-170 | \ | \ | 1-110 | 0.5-3.5 |
| Simulation 2 | cGENIE $O_2$. | 10-170 (1/k) | 0.01-1 | 20 | 1-110 | 0.5-3.5 |
| | WOA. | 10-170 (1/k) | 0.01-1 | 20 | 1-110 | 0.5-3.5 |
| Simulation 3 | cGENIE $O_2$ | \ | 0.01-1 | \ | 1-100 | 0.5-3.5 |
| | WOA | \ | 0.01-1 | \ | 1-100 | 0.5-3.5 |






## 3.    Results

In this section, we start by summarizing the overall statistical outcome of the tuning, then present a series of spatial analysis comparisons for each of the highest M-score ensemble members. The spatial analyses progressively reduce in scale, moving from global surface distributions, to global and basin-specific water column profiles, and finally to spatial comparisons for a specific ODZ region.

### 3.1.    Model skill score

The M-score values achieved across the complete ensemble for each of the 3 main parameterization-combinations are illustrated in Fig. 3. The sensitivity test shows the M scores are sensitive to all three of the main parameters for the ensembles in Fig. 3. Higher model skill scores are usually reached when "threshold" is tuned to 10 μM [$O_2$] for all the ensembles, including both model-simulated [$O_2$] and WOA-forced [$O_2$]. For the ensembles, "lifetime-threshold" and "Fennel-threshold", the highest M scores are similar—0.305 and 0.308, respectively (Table 2). Both these ensembles have the highest performance when "threshold", "lifetime", and I:C ratio are tuned to 10 μM [$O_2$], 50 years, and 1.5 × 10$^{-4}$ mol/mol, respectively, which is generally consistent with observations (Lu et al., 2016, 2020b; Tsunogai, 1971; Elderfield and Truesdale, 1980) (discussed in more detail later). The model performance of "reminO2lifetime-threshold" is less good than the other two combinations, with the best M score of 0.266 when "threshold", "reminO2lifetime", and I:C ratio are tuned to 10 μM $O_2$, 1 × 10$^{-6}$ mol/kg, and 3.5 × 10$^{-4}$ mol/mol, respectively (Table 2, Fig. 3). We note that for each parameterization-combination, the highest possible M score achievable by tuning improves when [$O_2$] is forced to that of the World Ocean Atlas 18 (WOA18) climatology (Garcia et al., 2018).

### 3.2.    Meridional surface I$^-$ distribution

Figure 4 shows a comparison between the observed latitudinal distribution of [I$^-$] at the surface and as simulated by the model for each parameterization-combination (for the respective best M-score ensemble member). Note that the observations (Section 2.3) are binned to the corresponding model grid cells and as such, reflect averages over the upper-most 80 m of the water column. This represents a reduction from 1338 to 141 surface ocean data points. We find that the surface ocean [I$^-$] in the model shows a trend of increasing values towards low latitudes, broadly consistent with observations (Chance et al., 2014) (Fig. 4). The "lifetime-threshold" and "Fennel-threshold" show similar latitudinal trends, but both overestimate the surface I$^-$ in the mid-low latitudes in the southern hemisphere. The "reminO2lifetime-threshold" ensemble produces better estimation of meridional surface [I$^-$] trend, although overestimates [I$^-$] in the tropical surface ocean compared to the other two ensembles (Fig. 4).



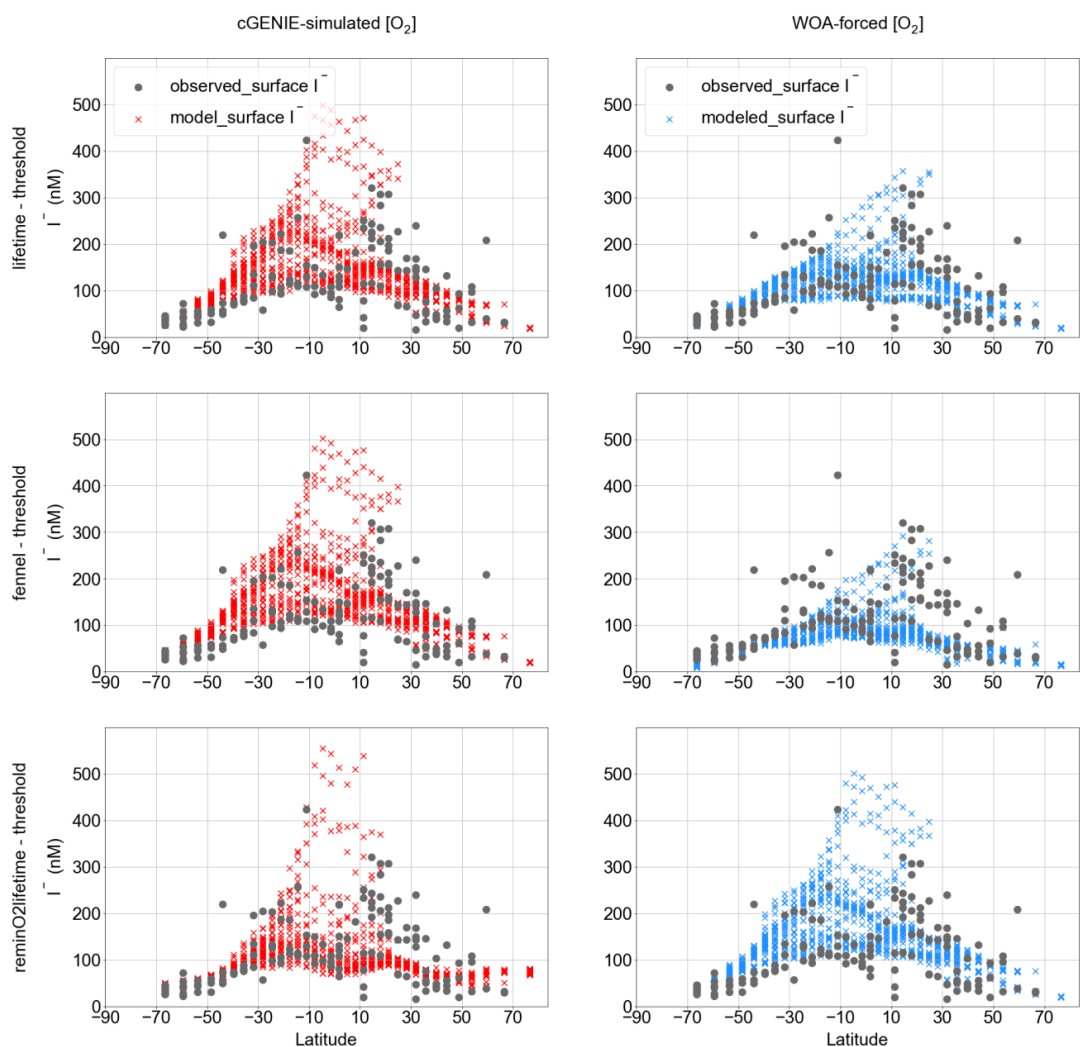

**Figure 4: Modeled latitudinal surface iodide distribution compared with observation with the cGENIE simulated [O₂] and the [O₂] restoring forcing. The elevated [I⁻] observed and modeled in low latitudes is the result of phytoplankton reduction in the surface ocean. Note that the I⁻ distribution simulated by "lifetime-threshold" and "fennel-threshold" are close but not identical.**

## 3.3. Global and basin-specific iodine depth distributions

Comparisons between the observed distributions of I⁻ and IO₃⁻ seawater concentrations among the global ocean and the Atlantic and Pacific Oceans are presented in Fig. 5. Again, we re-gridded the iodine observations (see: Section 2.4) and selected sub-sets of the data that lay in either Atlantic or Pacific basins, contrasting with the corresponding model values at those locations. We find only relatively minor differences between the best M-score ensemble member of



each of all three parameterization-combinations, and all show increased [IO$_3^-$] and decreased [I$^-$] with increased depth

below the photic zone in the Atlantic and Pacific basins, as well as globally (Fig. 5). The modeled depth profile broadly

matches with observation in the Atlantic and deep Pacific Ocean, except the underestimated subsurface peak of [I$^-$]

observed in the Pacific and overestimated [IO$_3^-$] in the deep Pacific (Fig. 5). This mismatch of subsurface I$^-$ peak is

probably the result of sampling bias, with most of the Pacific iodine observations from ODZs in the Eastern Tropical

North Pacific (ETNP) and the Eastern Tropical South Pacific (ETSP). For example, in model depth profiles masked

to only include grid points with corresponding observations, the modelled Pacific depth profiles show a clear mid-

depth ODZ feature (Fig. S8).

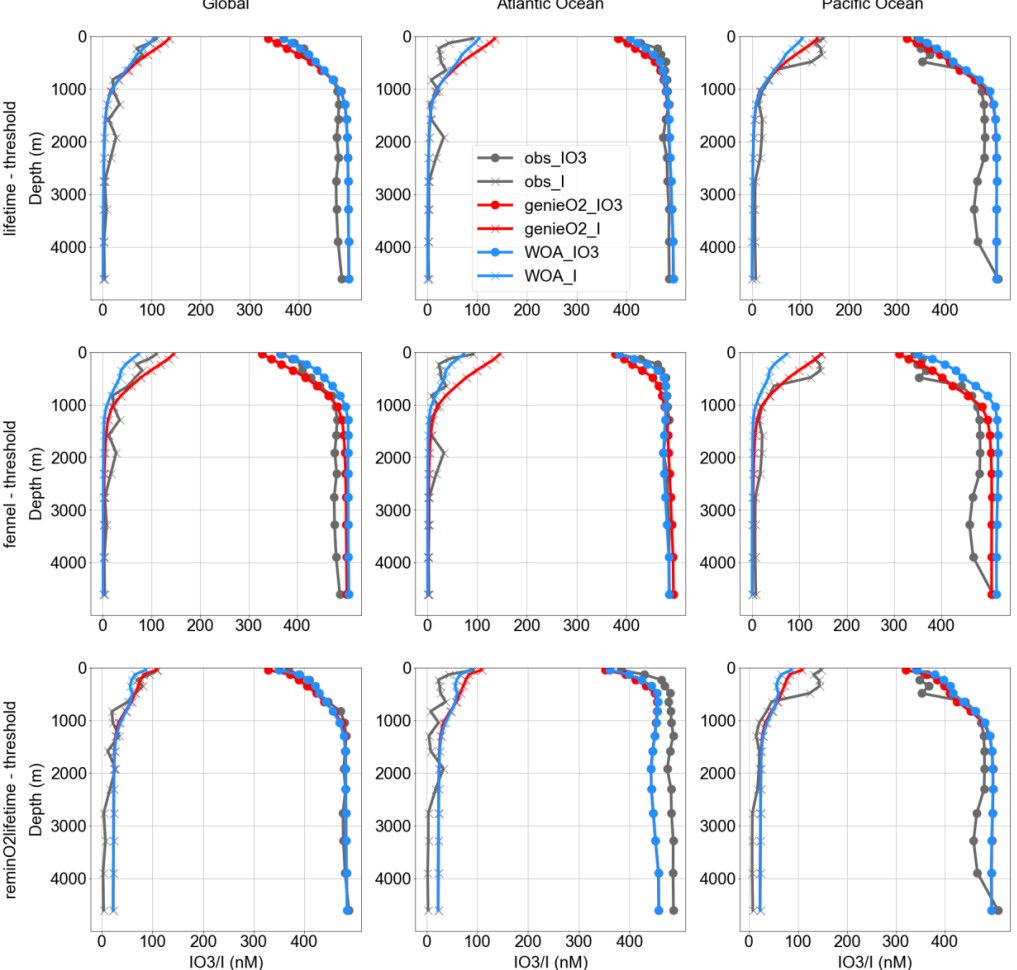

**Figure 5: Modeled averaged iodine (including iodate and iodide) depth profile among global ocean, the Pacific, the Atlantic compared with observation. The surface I$^-$ enrichment among the ocean basins is caused by phytoplankton reduction. The subsurface (~500m) I$^-$ enrichment is the result of sampling bias that most of the observations are from the ETNP and ETSP ODZs.**






### 3.4.    Iodine distribution within ODZs

To assess the model ability to simulate iodine cycling in marine low oxygen environments, we compared distributions
of oxygen and iodine species in the ETNP (Fig. 6). The $O_2$ transects amongst all model simulations are the same
because we only changed the parameterizations of the iodine cycle between ensembles and ensemble members (i.e.,
they all simulate the same biological pump in the ocean). Notably, compared to $[O_2]$ measured in the ETNP transect,
the model underestimates the extent of the ODZ. Severe deoxygenation below 50µM $[O_2]$ was observed in relatively
shallow depths between 100-200m in the ETNP, and this ODZ extends for more than 3000 km towards off-shore from
Mexican coast (Fig. 6). Although cGENIE simulates the $O_2$-deficient pattern in the ETNP, the extent of the ODZ is
underestimated. The simulated oxycline is ~200m deeper than the observation and the $[O_2]$ variation is gradual. The
ODZ below 20µM $[O_2]$ in the model is limited to a small spatial extent within 1000km offshore, which is much smaller
than that in the observation (Fig. 6).

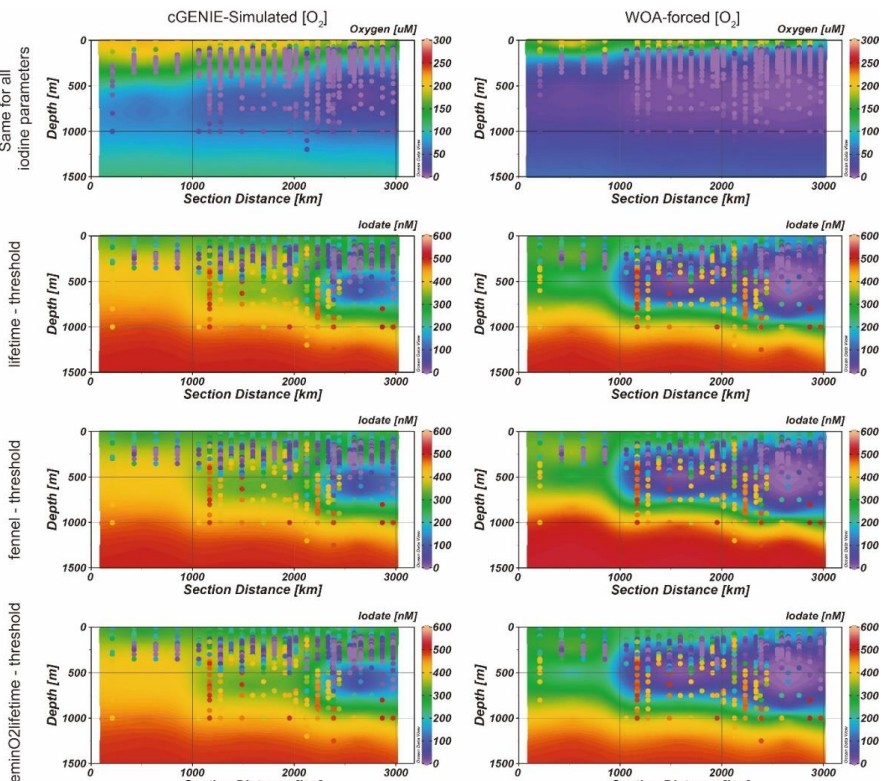

**Figure 6: Modeled (contour) and observed (colored dots) west-to-east transect of $IO_3^-$ and $O_2$ in the ETNP. Note that the WOA-forced $[O_2]$ models simulate a larger extent of $IO_3^-$ anomaly, which better matches the observation. The left-hand panel contours are model results based on cGENIE-simulated $[O_2]$ while contours on the right are model results from WOA-forced $[O_2]$.**






All the three chosen best-performance-experiments show similar iodine anomalies ($IO_3^-$ depletion) in the
ETNP, fitting the general feature of the observation. Other parameterizations did not replicate the ODZ (Fig. S4).
However, even under the "best-fitting" parameters, compare to the observations, the ODZ feature in the model is
underestimated both in intensity and in areal extent compared to the observations (Fig. 6). The observed $IO_3^-$ depletion
zone ($[IO_3^-]$ <100nM) occurs in shallower depths between 100-500m, matching the shallow ODZ and spans ~2000km
offshore; however, the modeled $IO_3^-$ depletion zones in the ETNP are located in 400-700m, and only extends within
1000km from the shore.
We also ran model ensembles forcing cGENIE to restore the modern ocean $[O_2]$ annual average climatology
to that of the WOA18 (Garcia et al., 2018) (Fig. 6). Now the subsurface $IO_3^-$ depletion zone in the ETNP ODZ in all
three ensembles extends ~2000km offshore and spans across 100-1000m in depth (Fig. 6). The shallow and extended
ODZ iodine distribution in the ETNP better matches the observation compared to non-$O_2$ restoration simulations.

**Table 2. The performance of the cGENIE iodine simulations and associated parameterization when the**
**model reaches the best global M score. Note that the oxidation rate constant *k* in 'Fennel' is in unit of year$^{-1}$ in**
**the model configuration, which is also the reciprocal of the 'lifetime'. The full model performance is**
**summarized in Table S2. Note that the lifetime in 'Fennel' is parameterized as k (year$^{-1}$) = 1/lifetime.**

| Parameter description | | Iodine oxidation parameters | | | Iodine reduction parameters | I:C ratio (× $10^{-4}$ mol/mol) | Model skill score |
|---|---|---|---|---|---|---|---|
| | | 'lifetime' (years) | 'reminO2lifetime' (× $10^{-5}$ mol/kg) | 'Fennel' (Inhibition constant/ μM $O_2$) | 'threshold' (μM $O_2$) | | |
| Simulation 1 | cGENIE $O_2$ | 50 | \ | \ | 10 | 1.5 | 0.305 |
| | WOA | 50 | \ | \ | 10 | 1.5 | 0.385 |
| Simulation 2 | cGENIE $O_2$ | 50 (1/k) | \ | 20 | 10 | 1.5 | 0.308 |
| | WOA | 10 (1/k) | \ | 20 | 10 | 3.5 | 0.385 |
| Simulation 3 | cGENIE $O_2$ | \ | 0.1 | \ | 10 | 3.5 | 0.266 |
| | WOA | \ | 0.1 | \ | 10 | 3.5 | 0.365 |




**4.    Discussion**

In summary: we ran ensembles for five combinations of iodine cycling parameters (summarized in Table S2) in the
cGENIE Earth system model (both with internally calculated and WOA-imposed [$O_2$] distribution) and presented
the results for 3 of them that showed the best performance. We analyzed the performance of the ensembles via: (1)
M score for the model-data match of both [$I^-$] and [$IO_3^-$] across the entire ocean, (2) qualitative model-observation
comparison of latitudinal surface ocean distributions of [$I^-$], (3) averaged depth profiles in global and individual
ocean basins for both [$I^-$] and [$IO_3^-$],  (4) iodine transects across the across the Eastern Tropical North Pacific
(ETNP) ODZ, and (5) M score for model and I:Ca observations for pre-OAE2.

**4.1.    Overall model skill score comparison**

Two broad observations emerge from the M score comparison. First, the 1$^{st}$-order kinetic iodine oxidation associated
ensembles ("lifetime-threshold" and "Fennel-threshold") have the highest M scores (Table 2), under both cGENIE-
simulated [$O_2$] and WOA-forced [$O_2$]. This is consistent with previous observations of 1$^{st}$-order kinetics for
I$^-$  oxidation (Tsunogai, 1971). Second, the simulations with WOA-forced [$O_2$] produce significantly higher M
scores than that of the cGENIE-simulated [$O_2$] field (at least ~0.8 of improvement; Table 2). Despite a 1$^{st}$-order non-
$O_2$ dependent oxidation mechanism providing the highest M scores, the WOA vs internally model-generated
dissolved oxygen distributions comparison highlights the $O_2$ and related redox dependency of the iodine cycle from
the perspective of $IO_3^-$ reduction. Each of these factors are discussed in the following Section 4.1.1.

**4.1.1.    Parameter value plausibility**

A credible representation of the marine iodine cycle requires not only that observations can be replicated, but that
replication occurs when parameter values fall within real-world ranges. In this section, we discuss the validity of our
best-fit (maximized M-score) parameter values. For the iodine cycle, these parameters include $O_2$ threshold, I:C ratio,
and I$^-$ oxidation rate.
Our model M score is highest with an [$O_2$] reduction threshold of 10μM (Fig. 3 and Table 2). Although it is
generally well accepted that $IO_3^-$ is reduced in low oxygen settings (Luther, 1991; Rue et al., 1997; Wong et al., 1985;
Wong and Brewer, 1977; Farrenkopf and Luther, 2002), the degree of $O_2$ depletion that triggers $IO_3^-$ reduction is still
unclear. A relative high threshold for triggering $IO_3^-$ reduction is proposed based on comparison between benthic
foraminiferal I:Ca and ambient [$O_2$] (20-70 μM $O_2$; Lu et al., 2016, 2020a). These [$O_2$] thresholds are similar to that
determined in a previous cGENIE-based iodine cycle study (30 μM) (Lu et al., 2018), but it is difficult to directly
compare this to our results because of differences in the model representation of the ocean biological pump, the iodine
observational data-set, and model-data comparison methods utilized.
Many of the studies suggesting a relatively high [$O_2$] threshold is based on evaluations of [$IO_3^-$]-[$O_2$] within
the upper chemocline of ODZs; however, evaluation of [$O_2$]-[$IO_3^-$] from ODZs throughout the entire water column
suggest the potential for $IO_3^-$ persistence within the low oxygen cores of ODZs. Specifically, $IO_3^-$ accumulation is
observed within the ETNP and ETSP at depths where [$O_2$] is close to or below the detection limit of the sensors which
is reported near 1 μM (Hardisty et al., 2021). In addition, it is worth noting that the kinetics of $IO_3^-$ reduction is
heterogeneous both within and between ODZs. For example, a transect evaluating $IO_3^-$ reduction rates in the ETNP



observed rapid rates in the upper oxycline, where [$O_2$] was near ~12 µM, but the potential for sluggish rates in the
ODZ cores, where [$O_2$] was below detection. In an early study of the Arabian Sea, $IO_3^-$ was reduced rapidly within
the ODZ core. Together, these suggest $IO_3^-$ reduction may be controlled by factors beyond $O_2$ (Hardisty et al., 2021;
Farrenkopf and Luther, 2002). For example, $IO_3^-$ is likely formed in high [$O_2$], non-ODZ water masses but can be
sustained upon transport or mixing within oligotrophic, offshore ODZ regions where organic supplies are more limited
(Hardisty et al., 2021). A comparison to the N cycle would also indicate a low [$O_2$] threshold—for example,
denitrification has a sub-µM [$O_2$] threshold and has a similar redox potential with $IO_3^-$ reduction (Dalsgaard et al.,
2014; Thamdrup et al., 2012). A sub-µM [$O_2$] threshold for $IO_3^-$ reduction could explain the [$IO_3^-$] variations observed
in ODZ cores with [$O_2$] below the µM detection limits of sensors; however, iodine speciation has yet to be analyzed
alongside [$O_2$] measurements via sensors with sub-µM detection limits, such as STOX sensors. Regardless, our 10
µM [$O_2$] threshold based on maximizing the M score reflects a global average value and clearly falls within the ranges
of oceanographic observations.

For both our study and that of Lu et al., (2018), an $I^-$ lifetime of 50 years maximizes model performance.

However, Lu et al., (2018) chose to implement a slightly lower value of 40 years for their paleo-application because
it reflected the slowest rate observed in the literature at that time (Tsunogai, 1971). Notably, though $IO_3^-$ is the most
abundant marine iodine species, its production from $I^-$ has never been observed under normal marine conditions. This
has acted as a major hinderance on providing direct constraints. That said, our cGENIE estimate is consistent with a
multitude of other constraints that indicate that $I^-$ oxidation to $IO_3^-$ undergoes extremely slow kinetics. The $I^-$ oxidation
rates calculated through indirect methods including mass balance and seasonal iodine speciation changes (Tsunogai,
1971; Campos et al., 1996; Truesdale et al., 2001; He et al., 2013; Edwards and Truesdale, 1997; Žic et al., 2013;
Moriyasu et al., 2023) or through radiogenic tracer spiked incubations (Hardisty et al., 2020; Schnur et al., 2024;
Ștreangă et al., 2024) have a wide range of variation from 1.5nM/yr to 670nM/yr. The lifetime in cGENIE is 50 years,
which can be approximately converted to the zeroth order rate of <9 nM/yr, falling in the lower end of the previous
studies.

Our best-fit I:C ratio is $1.5 \times 10^{-4}$, and this value is in agreement with plankton measurements and mass-

balance calculations (Chance et al., 2010; Elderfield and Truesdale, 1980). In the photic zone, $IO_3^-$ is taken up by
phytoplankton and incorporated into their cellular structures followed by subsequent conversion into $I^-$ (Hepach et al.,
2020). Due to this, it is assumed that $IO_3^-$ removal in the surface layer of the ocean is a function of organic carbon
fixation during primary productivity according to Redfield-like ratios (Campos et al., 1996; Chance et al., 2010). Of
the parameters incorporated into the model, in theory, I:C should probably be the best constrained. However, published
I:C ratios based on field and laboratory measurements vary over several orders of magnitude ($10^{-5}$ to $10^{-3}$) (Elderfield
and Truesdale, 1980; Campos et al., 1996; Hepach et al., 2020; Chance et al., 2010). To limit the number of model
simulations and size of the ensembles, we only varied I:C between $5 \times 10^{-5}$ and $3.5 \times 10^{-4}$, increasing by $1.0 \times 10^{-4}$, which
covers the range indicated by previous studies (Elderfield and Truesdale, 1980).
It is unlikely that the I:C value is constant across the global ocean due to differences in phytoplankton compositions
and other factors. In cGENIE, most of the elevated surface [$I^-$] over 200nM is present in the ETSP and the northern
Indian Ocean, representing the effect of high primary productivity and transformation of $IO_3^-$ to $I^-$ via the rapid



recycling DOM 'loss term' in the representation of biological export (Fig. S6). The mismatch between the model and
the observation probably hints that the I:C ratio is not constant in the ocean, as which is also hypothesized by Wadley
et al., (2020), although in the absence of an explicit representation of primary productivity in the model and lack of a
spatially variable f-ratio (Laws et al., 2000)(implicitly, the f-ratio is 0.33 everywhere in cGENIE). In testing a fixed,
spatially uniform I:C, Wadley et al. (2020) underestimated surface [I⁻] in low latitudes and overestimated
concentrations in mid-latitudes. Based on their model-observation comparison, they hypothesized that the I:C ratio
decreases systematically with sea surface temperature (SST) (Wadley et al., 2020). Until more constraints are
developed on spatial variability and associated driving factors for I:C, a generalized approach of a globally uniform
I:C remains the most parsimonious assumption, especially considering cGENIE's intended extrapolation to ancient
settings.

### 4.1.2.    Comparing alternative iodide oxidation parameterizations in cGENIE

Due to similarity in redox potentials, the iodine cycling in the ocean has been hypothesized to be linked to the cycling
of nitrogen (Rue et al., 1997). Nitrification promoting I⁻ oxidation to $IO_3^-$ has been hypothesized in field studies
(Truesdale et al., 2001; Žic et al., 2013), and more recently has been linked via observation of I⁻ oxidation to $IO_3^-$ by
ammonia oxidizing bacteria in laboratory environments (Hughes et al., 2021). We further note that Wadley et al.,
(2020) explicitly link I⁻ oxidation to $NH_4^+$ oxidation in their surface ocean iodine cycle model.

As an alternative to the first-order lifetime oxidation parameterization used here and in Lu et al., (2018) and

in the current absence of a full nitrogen cycle (and hence explicit $NH_4^+$ oxidation) in cGENIE, we also applied a
strategy ("reminO2lifetime") which links I⁻ oxidation to organic carbon remineralization and the consumption rate of
dissolved oxygen. The reasoning behind this is that the $O_2$ consumption rate in the model implicitly reflects bacterial
oxidizing activity in the water column and hence the potential for I⁻ to be oxidized to $IO_3^-$.

We find that the overall model performance involved with the "reminO2lifetime" is lower than other

experiments where I⁻ oxidation is ubiquitously oxidized according to a parametrized lifetime, or "lifetime-threshold"
(maximum M score 0.266 vs. 0.305/0.308 under cGENIE simulated [$O_2$]) (Fig. 3 and Table 2). However, despite
slightly lower M scores, the "reminO2lifetime" generally replicates the latitudinal surface [I⁻] trend, the depth profiles
in the ocean basins, and the ODZ transect (Figs. 4-6).

Under the default "lifetime" scheme, I⁻ will oxidize in the whole ocean regardless of the concentration (or

even presence/absence) of ambient $O_2$. This scenario might hence not perform well in replicating the ocean iodine
cycling at intervals during the Phanerozoic when ODZs were highly expanded, or particularly during the Precambrian
when the majority of the ocean was ferruginous or euxinic and highly depleted in $O_2$ (for example, summarized by
Lyons et al., 2014). Ideally, for application to paleoceanographic studies, an [$O_2$]-related I⁻ oxidation alternate scheme
is required. Although thermodynamics theory suggests $O_2$ does not directly drive I⁻ oxidation (Luther et al., 1995),
field studies in ODZs indeed observed low [$O_2$] inhibits this process (Farrenkopf and Luther, 2002; Moriyasu et al.,
2020). We hence developed and tested variable I⁻ oxidation kinetics, with the ambient dissolved $O_2$ providing an
inhibition of the rate of oxidation based on the enzymatic nitrification scheme of Fennel et al., (2005).





Since most of the ocean is fully oxygenated today, the difference of M scores between "lifetime" and "Fennel"

oxidation models are minor (0.305 vs. 0.308, Table 2). The parameters associated with the highest M score between
two oxidation options are also very close to each other, except "Fennel" oxidation together with the model WOA-
forcing, has higher I:C ratio ($3.5 \times 10^{-4}$) and faster saturated I$^-$ oxidation kinetics (0.1 yr$^{-1}$ vs. 0.02 yr$^{-1}$ in other
ensembles). The parameter differences between the "Fennel"-WOA ensemble and other models make sense because
the faster oxidation rate compensates the increased I$^-$ production through the higher biotic uptake rate. The pre-OAE2
simulations are particularly illustrative of this tradeoff and are discussed in more detail in section 4.3.

In summary, all the three parameter combinations produce high and comparable M scores and similar

parameters (oxidation, reduction, and I:C) associated with these M scores (Table 2). Although direct field-based
evidence to evaluate the controlling parameters of "reminO2lifetime" is absent, the parameters controlling the other
model scheme are consistent with previous studies.
**4.2.**      **Beyond the M-score: model-data comparison across iodine gradients**
As applied here, the M score provides a quantitative measure that describes the overall model global performance and
allows us to directly compare the implications of parameter value and parameterization choices. However, the M score
overlooks regional gradients that may be important for both paleo- and modern oceanographic research. Indeed,
amongst all the various ensembles we ran as part of this study (Table S1), only "lifetime-threshold", "lifetime-Fennel",
and "reminO2lifetime-threshold" performed well in replicating the modern oceanic iodine gradients (Figs. S2-S4) and
informed our decision to focus on these 3 parameterization-combinations here. We now discuss this in more detail
below.
**4.2.1.**      **Meridional surface [I$^-$] gradient**
All the parameterization-combinations summarized in Table 2, as well as the observations, show enrichment of I$^-$ in
the surface ocean at low latitudes (Fig. 4). The pathway of transforming IO$_3^-$ into I$^-$ in these oxidized waters is through
primary productivity in the photic zone, which is temperature dependent (Chance et al., 2014). A recent north-south
transect showed the highest surface I$^-$ enrichment in the oligotrophic, permanently stratified tropical stations (Moriyasu
et al., 2023). Therefore, the IO$_3^-$ flux from deep waters through seasonal mixing may be an important balance to *in-*
*situ* IO$_3^-$ reduction rate by primary producers in the high latitudes (Chance et al., 2014; Moriyasu et al., 2023). The
cGENIE model generates the general pattern of latitudinal surface I$^-$ distribution pattern; however, overestimation
especially in low latitudes may exist, especially in the tropical surface where [I$^-$] are close to 500nM among all the
cGENIE-O$_2$ models (Fig. 4).

The cGENIE (internally generated oxygen distributions) vs WOA (imposed distributions) O$_2$ comparison

provides evidence that I$^-$ generated in low [O$_2$] settings may broadly enhance [I$^-$] in oxygenated photic waters, with
lower and closer-to-observations [I$^-$] values in the WOA tunings (Fig. 4). This includes "lifetime-threshold", where
O$_2$ only impacts the reductive portion of the iodine cycle, but also the "Fennel" and "reminO2lifetime" where rates of
I$^-$ oxidation is also [O$_2$] dependent. More specifically, most of the elevated (over 200nM) surface [I$^-$] in cGENIE,
occurs in the ETSP and the northern Indian Ocean and corresponds to locations of high primary productivity (Fig. S6).





Since the surface ocean [O₂] in the model is never below 200μM, IO₃⁻ reduction at the ocean surface is unlikely.
Instead, *ex situ* transport from proximal regions of subsurface anoxia is the most probable source of elevated I⁻. Indeed,
the most prominent regions of I⁻ enrichment in the model occur near the Peruvian coast and in the Arabian Sea, where
ODZs lie below the surface (Fig. S6).  More detailed data-model comparison among these two areas is limited because
the observation data are few (e.g., Farrenkopf and Luther, 2002 and Rapp et al., 2020). In contrast, the meridional
trend of I⁻ in the surface Atlantic Ocean, where ODZs are less developed, exhibits better agreement with both the
observation and the Wadley et al., (2020) model (Fig. S6). The overestimation of tropical ocean surface [I⁻] by cGENIE
is hence likely to be associated with deficiencies in the simulation of ODZ oxygenation.
Importantly, modeled overestimations in surface ocean [I⁻] may be difficult to confirm given current
observational data densities. Specifically, a comparison of observational data to model-latitudinal trends masked to
only include grid points with corresponding observations show the same broad trend of increasing [I⁻] in the low
latitude but with fewer so called "overestimations" (Fig. S7). More observations in surface ocean [I⁻] from low
latitudes is required to better assess the validity of elevated modelled surface ocean [I⁻] in some regions.

### 4.2.2.    Global and basin-specific iodine depth profiles

All the models, both cGENIE-[O₂] and WOA-forced, generate a decrease in [I⁻] and increase in [IO₃⁻] from the
euphotic zone down to the deep abyssal zone across ocean basins, matching the primary-production-driven pattern
(Fig. 5). As discussed in the previous section, this surface maxima of [I⁻] in the oxygenated water column is the result
of biologically mediated reduction or release during cell senescence of phytoplankton. Below the photic zone, [I⁻] is
close to zero and IO₃⁻ becomes the dominant species. The deep ocean is mostly oxygenated and has longer water
residence times (several millennia, Matsumoto, (2007)) compared to the I⁻ lifetime (<40 years, Tsunogai, (1971)), thus
facilitating I⁻ oxidation in the absence of IO₃⁻ reduction in ODZs.
We note that there are multiple general discrepancies between observations and data as well as differences
between WOA and cGENIE-[O₂]. In general, all models reproduce the global average better, relative to the basin-
specific profiles. We suggest that the global averaged profiles are a better test of the cGENIE simulations because of
sampling biases associated with individual basins. For example, the discrepancy between the model and the
observation is prominent in the Pacific (Fig. 5). The observed Pacific subsurface [I⁻] peak mirrors the [IO₃⁻] minima
that occurs at depths where ODZs are present. This ODZ feature in the averaged Pacific observation profile is likely
the result of sampling bias since most of the observations from the Pacific were from the ETNP (Rue et al., 1997;
Moriyasu et al., 2020), thus not reflecting the overall iodine distribution in the Pacific Ocean (Fig. S8). As for the
meridional trends, sampling bias is again demonstrated in a depth profiles masked to only include modeled grid points
with corresponding observation data, with a clear mid-depth ODZ feature in the modeled Pacific depth profiles (Fig.
S8). A similar example is from the Indian Ocean, which we do not show, since most iodine subsurface data come from
the ODZ, not normal marine, localities. All this said, while the general features of iodine speciation with depth are
generally similar, our data compilation indicates the potential for some basin-specific variations which require further
research to validate and mechanistically understand.

### 4.2.3.    Iodine distribution within ODZs



One of the major goals of calibrating the iodine cycle in cGENIE is to simulate the iodine behavior associated with
ancient low oxygen settings. To assess this potential, we analyzed model performance for the ETNP (Rue et al., 1997;
Moriyasu et al., 2020) where observation data are abundant and the areal extent of the ODZ is sufficiently large to be
reflected in the model grid (Fig. 6). Importantly, it should still be noted that cGENIE is best applied to understanding
broad scale processes and thus the scope of the ETNP ODZ transect comparison is likely too fine resolution to expect
a good match. That said, the simulated reduction in $IO_3^-$ to $I^-$ generally overlaps with the extent of the ODZ (Fig. 6),
which provides support for the use of cGENIE to understand the broad distribution of ancient $[IO_3^-]$ and $[O_2]$. Other
non-threshold model parameterization-combinations (shown in Fig. S4) did not replicate the ODZ feature in iodine
speciation.
Across all model configurations assuming cGENIE-$[O_2]$, the prominent discrepancy is an underestimation of
the spatial extent and intensity of the $IO_3^-$ depletion zone in the ETNP (Fig. 6) because simulated subsurface $O_2$
deficient area is notably narrower than that compared to WOA climatology (Fig. 6, Fig. S5). There are multiple factors
that might affect the performance of simulating the $O_2$ cycle in cGENIE. The model might underestimate the intensity
of gyres in the North and Tropical Pacific, resulting in the narrowed ODZs in these areas. Also, the pattern of upwelled
nutrients into the surficial Tropical Pacific needs to be tuned to better replicate the productivity and the $O_2$ consumption
during remineralization. Another source of uncertainty is that the short-term processes, such as seasonal or El Niño
driven ODZ variations at the time of sample collection are not replicated in the cGENIE simulations.
Other data-model misfits might have arisen as a result of additional $IO_3^-$-reduction dependencies not
explicitly accounted for in the model. As discussed above, shipboard incubation and radiogenic-tracer-spiked rate
calculation suggest that $IO_3^-$ reduction is slow in the offshore ETNP ODZ (Hardisty et al., 2021). This could explain
why measurable $IO_3^-$ is present in the core of the ETNP ODZ (Fig. 6). This is further exacerbated in the Eastern
Tropical South Pacific ODZ, where $[IO_3^-]$ remains above 250 nM in some studies (Cutter et al., 2018) but near
detection limits in in others (Rapp et al., 2020), suggesting extreme spatiotemporal variability related to currently
unconstrained factors. Further, while the capability of microbes using $IO_3^-$ as an electron acceptor for oxidizing organic
matter has been confirmed in laboratory culture experiments (Councell et al., 1997; Reyes-Umana et al., 2021;
Yamazaki et al., 2020; Amachi et al., 2007; Farrenkopf et al., 1997), no study to date has demonstrated non-$O_2$
dependent controls driving variable $IO_3^-$ reduction rates.
An important factor contributing to elevated $[I^-]$ in ODZs is benthic fluxes and reduction occurring within
the uppermost layers of marine sediments (akin to denitrification). To help account for this in our M score and model
calibration (see: methods section), excess iodine was filtered from our observational dataset. The excess iodine
originated from the sediment flux has been observed in ODZ water columns contacting anoxic sediments (Chapman,
1983; Farrenkopf and Luther, 2002; Cutter et al., 2018; Moriyasu et al., 2020). We note that excess iodine is explicitly
as $I^-$, reflecting the limited or lack of oxygen within the ODZ, and is a local-regional phenomenon not yet observed to
persist beyond ODZ settings. As such, since our goal is a paleoceanography-focused model, cGENIE does not
incorporate the benthic flux of $I^-$ because only $IO_3^-$ is tracked via the I:Ca paleoredox proxy.

**4.3.     Applicability of the cGENIE marine iodine cycle to paleo redox reconstruction**



Parameter tuning, and the ability to reproduce modern observations, does not by itself offer any guarantee that spatial
patterns are being simulated for the 'correct' reason (i.e., specific set and relative importance of mechanisms). This is
even more pertinent in the context of the application of a modern-tuned model to paleo redox reconstruction. To test
whether our new iodine cycle had predictive power in the geological past, we carried out a deep-time plausibility test.
For the paleo plausibility test, we adopted the Cretaceous, pre-OAE2 (ca. 93 Ma) configuration (continental
arrangement and ocean bathymetry, wind stress and velocity, and zonal average planetary albedo boundary conditions)
from Monteiro et al., (2012). We choose this particular geological interval because the controls on ocean redox have
been previously evaluated using the GENIE model (Monteiro et al., 2012; Hülse et al., 2019), the oceanic conditions
are much more extensively dysoxic and anoxic than present-day and hence represent a relatively severe test of the
model iodine cycle, and  a number of I:Ca proxy measurements are available (Zhou et al., 2015). In order to evaluate
the same configuration of the iodine cycle as optimized in this study, we also substituted the temperature-independent
representation of biological export production and fixed remineralization profile of POM in the water column (i.e.,
Ridgwell et al., (2007)) for the temperature-dependent scheme of Crichton et al., (2021). However, in substituting the
biological pump scheme in the model we change the ocean redox landscape compared with e.g., Monteiro et al., (2012).
We therefore test a range of different assumptions regarding the ocean $PO_4$ inventory at the time as a means of
generating a range of different plausible states of ocean oxygenation. In this, we test: 0.2, 0.4, 0.6, 0.8, 1.0, and 1.5
times the mean modern concentration (2.15 mM). We run the model with each of the best-fit (highest M-score) sets
of parameter values associated with the 5 different parameterization-combinations (but focus on the results of the same
3 combinations we did for comparison against modern), and for each of the varying $PO_4$ inventory assumptions, for
10,000 years to steady-state.

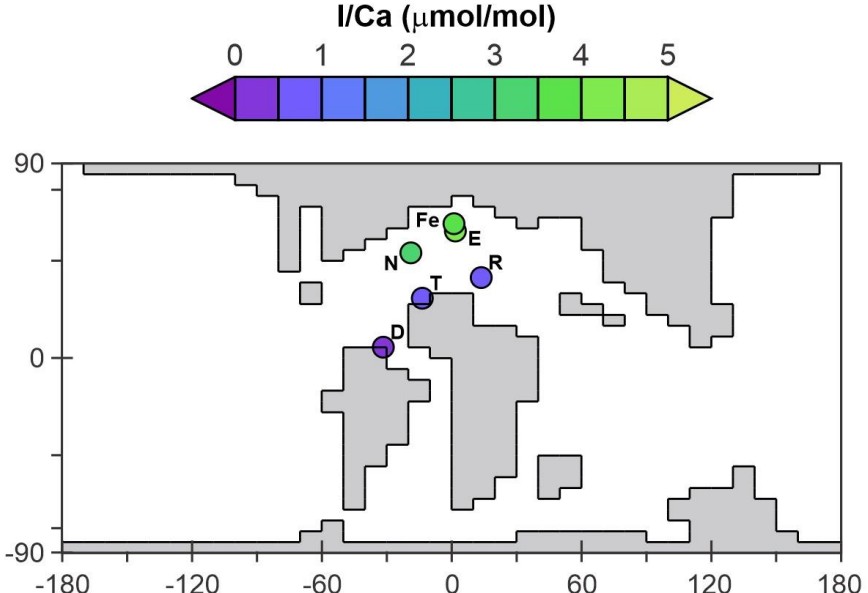


**Figure 7: The continental setting during the Cretaceous OAE2 (Cenomanian - Turonian) in cGENIE. The colored dots represent averaged pre-OAE2 I:Ca measurements from each of the sections. D = Demerara Rise; E = Eastbourne; Fe = South Ferriby; N = Newfoundland; R = Raia del Pedale; T = Tarfaya.**


The I:Ca data used for comparison with the model come from 6 sections (Zhou et al., 2015, Fig. 7, listed in

Table S3). The pre-OAE2 I:Ca baseline value from each section is estimated through averaging the pre-CIE I:Ca

measurements from Table S1 of Zhou et al., (2015). For quantitative comparison between the model and the I:Ca data,

we create an empirically derived forward proxy model for I:Ca. In this, we took the simulated concentration of $IO_3^-$

and $Ca^{2+}$ in the ocean surface layer of the model at every ocean grid point, and applied the temperature-dependent

linear incorporation relationship derived from inorganic calcite synthesis experiment of Zhou et al., (2014), to estimate

I:Ca. The converted modelled I:Ca at each section were then directly compared with the mean pre-OAE2 I:Ca

measurements using the M score (Fig. 8). We then extracted simulated I:Ca values from the model grid points

corresponding to the sections reported by Zhou et al., (2015) and calculated the M-score. The statistical results of this

comparison are illustrated in Fig. 8 for the 3 parameterizations chosen for focus in the main text and for Fig. S9 for

the full parameterization-combinations and for parameter calibrations derived from internally and WOA-forced

dissolved oxygen distributions.

602

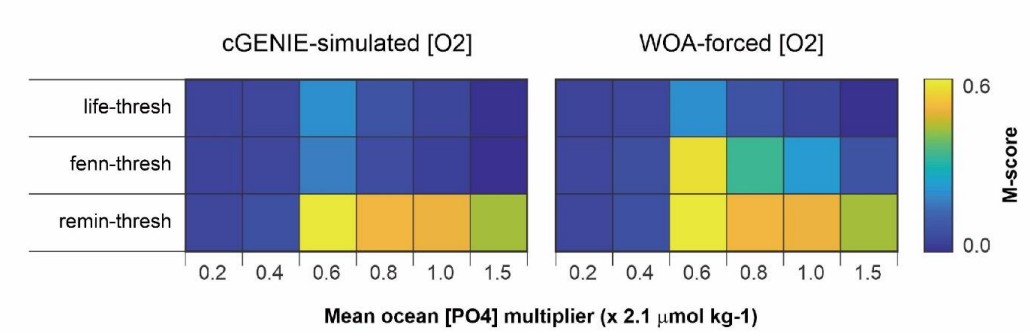

603

**Figure 8. The model skill scores of modeled and measured I:Ca during the pre-OAE2. The iodine cycling parameters are derived from modern simulations with cGENIE-simulated [$O_2$] and WOA-forced [$O_2$], respectively. lifetime-thresh = lifetime-threshold; remin-thresh +DOC = reminO2lifetime-threshold +DOC remineralization; remin-thresh = reminO2lifetime-threshold; fenn-thresh = fennel-threshold; life-inhib = lifetime-inhibition; life-remin +DOC = lifetime-reminSO4lifetime +DOC remineralization; life-remin = lifetime-reminSO4lifetime.**

610

Most of the parameterization-combinations tested reach their highest M scores under the assumption of 0.6-0.8 × modern [$PO_4$] (Fig. S9). Previous analysis using the same climatological configuration of the GENIE model indicated a $PO_4$ inventory of 1.0 x modern was most consistent with geological redox-related observations prior to OAE2 (Monteiro et al., 2012). However, our assumption here of temperature-dependent POM export and remineralization leads to higher export and shallower more intense ODZs compared to temperature-invariant biological scheme (see: Crichton et al., (2021)). Hence, for a similar degree of ocean anoxia, we would expect the need for a slightly lower nutrient inventory, as we indeed find.

In terms of the I:Ca M-score, we find the parameterization-combinations "reminO2lifetime-threshold" and "fennel-threshold" better replicate the geological observations compared to the "lifetime-threshold". In general, WOA-derived parameter sets perform better than those derived from cGENIE-[$O_2$], again hinting at the importance of reducing the uncertainties in simulating the modern $O_2$ cycle in cGENIE. These observations are also largely independent of the ocean $PO_4$ inventory assumption. Although combinations of parameterizations such as "reminO2lifetime-threshold" with DOC remineralization, "lifetime-inhibition", and "lifetime-reminSO4lifetime", also produce elevated M scores, these combinations do not perform well in replicating iodine gradients within the modern ocean (Figs. S2-4). Thus, until there is a better mechanistic understanding of $IO_3^-$ reduction in the modern ocean, the safest choice is arguably to apply the parameterization that best reproduces modern observations and hence retain use of the "threshold" $IO_3^-$ reduction parameterization for paleo applications.

The pre-OAE2 comparison is revealing because it encapsulates a strong gradient between high and very low I:Ca (Fig. 7), reflecting respectively, high and low surface ocean concentrations of $IO_3^-$ in the model. All three of the parameterization-combinations we focus on here (with WOA-derived parameter values) correctly lead to very low I:Ca values in the lower latitudinal sections (Demerara Rise, Tarfaya, and Raia del Pedale; Fig. 9), although with a tendency to slightly overestimate seawater $IO_3^-$ depletion (cross-plots in Fig. 9). Low ocean surface [$IO_3^-$] is due to



the existence of a circum-Equatorial band of intense sub-surface anoxia. In the higher latitudinal sections, including
Newfoundland, Eastbourne, and South Ferriby, I:Ca values tend to be underestimated to varying degrees (Fig. 9).
Compared to the "lifetime" parameterization, both "reminO2lifetime" and "fennel" oxidation simulate the I:Ca values
in these high latitudinal sections rather better, with the regression closer to the 1:1 line (dotted in Fig. 9). We find this
relationship instructional for understanding controls in the modern iodine cycle, which we discuss in more detail below.

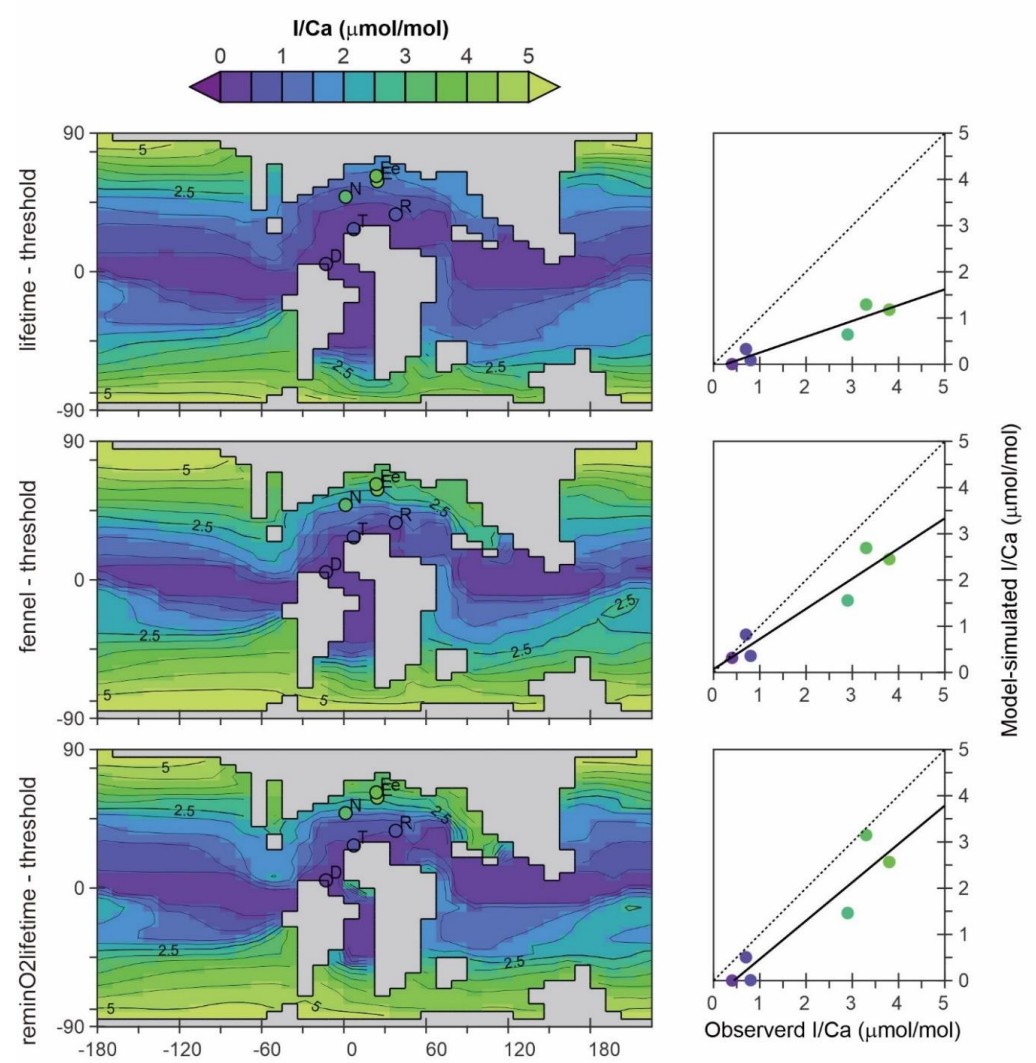


**Figure 9. The pre-OAE2 I:Ca field derived from cGENIE [IO₃⁻] simulations, and the comparison between**
**modeled and observed I:Ca from sections.**



We first note that both 'lifetime' and 'fennel' iodine oxidation parameterizations in conjunction with a
threshold of iodate reduction and internally generated GENIE-[$O_2$], give rise to identical parameter values (Table 2).
We infer that this is because the modern ocean is predominately well-oxygenated and hence there is little inhibition
of I$^-$ oxidation in practice. In the Cretaceous environment, although I$^-$ oxidation inhibition should be widespread, the
M-scores are similar (Fig. 8). The rate of I$^-$ oxidation in well oxygenated seawater is likely then critical in explaining
elevated I:Ca values at higher Cretaceous latitudes. However, simply decreasing the lifetime in the modern ocean
would result in an under estimation of surface ocean [I$^-$]. The 'fennel-threshold' combination under WOA-[$O_2$] reveals
a trade-off that solves this—a decreased I$^-$ lifetime compensated for by increased rates of I$^-$ release to the ocean interior
directly through the biological pump and elevated cellular I:C ($3.5 \times 10^{-4}$ vs. $1.5 \times 10^{-4}$). In the Cretaceous ocean this
combination allows for both sharper latitudinal gradients in [$IO_3^-$] (and hence I:Ca) to develop, as well as steeper
vertical gradients which allow for non-zero I:Ca values at low latitudes to be captured (cross-plot in Fig. 9). This slight
enhancement of the upper ocean [$IO_3^-$] gradient is also apparent in the present-day analysis (Fig. 5). The combination
of 'reminO2lifetime' with a reduction threshold works similarly—a shorter lifetime for I$^-$ under oxic conditions offset
in the modern ocean by elevated cellular I:C (Table 2). However, in this case, our gridded parameter search identifies
the trade-off as producing the highest M-score for both modelled and WOA oxygen distributions.
What we learn from this is that the cGENIE iodine cycle tuned to modern observations has predictive power
under a very different state of ocean oxygenation (and circulation and operation of biological pump). However, this
is not true for every choice of parameterization, and the simple 'lifetime-threshold' combination, which when
calibrated was statistically almost the best representation of the iodine cycle, was unable to reproduce the latitudinal
I:Ca gradients in the Cretaceous ocean. Shortening the lifetime (and adding an inhibition term) together with increasing
the assumed cellular I:C, could maintain modern ocean fidelity whilst much better capturing Cretaceous I:Ca. That
even better representations of Cretaceous I:Ca were possible but at the expense of reproducing modern observations
adequately hints that improvements in our mechanistic understanding are needed, although all of the above assumes
that the simulation of the Cretaceous redox landscape is plausible.
**5.    Conclusions**
Using the cGENIE Earth system model, we performed a series of ensemble experiment parameter searches for suitable
parameterizations to represent the marine iodine cycle and identified the best performing parameter value
combinations in each case. We found that the optimized parameters associated with $IO_3^-$ planktonic uptake, water
column $IO_3^-$ reduction and I$^-$ oxidation are within the range of field and experiment observations and hence plausible.
Three iodine cycling parameter combinations, "lifetime-threshold", "reminO2lifetime-threshold", and "fennel-
threshold" emerged as viable candidates following our tests of the global ocean model M score, and model-data
comparison across specific iodine gradients (euphotic latitudinal distribution, depth distribution, and ODZ
distribution). We further evaluate the plausibility of our parameterizations and their paleo and ocean deoxygenation
applicability by contrasting forward-proxy model generated I:Ca values against observations, taking the (pre-OAE2)
Cretaceous redox landscape as a case study. While some model-data discrepancies emerge for both modern and paleo,
these highlight that future observational and/or experimental work is necessary to better constrain modern iodine



cycling mechanisms and related spatiotemporal heterogeneities. While we further identified the importance of
improving the simulation of dissolved oxygen distributions in models, equally, we found that our conclusions
regarding preferred parameterizations and even specific parameter values, was not overly dependent on the specific
details of the simulated modern OMZs.  Overall, our analysis gives us a degree of confidence that carbonate I:Ca ratios
can be interpreted in terms of past ocean oxygenation using models such as 'cGENIE'.



*Competing interests. The contact author has declared that none of the authors has any competing interests.*

*Author contributions. KC, AR, and DH conceptualized the research presented in this paper. DH and AR acquired funding to support the study. AR developed iodine tracer and associated biogeochemical mechanisms in cGENIE. KC and DH designed model performance under the modern ocean configurations. KC compiled the modern ocean iodine database and the Cretaceous I:Ca data. KC ran the modern-ocean cGENIE analysis and performed model-data comparison. AR performed model-data evaluation for Cretaceous configurations. KC prepared the manuscript with contributions from all co-authors.*

*Acknowledgements. Funding support for DH and KC comes from NSF OCE 1829406. AR acknowledges support from National Science Foundation grant EAR-2121165 and the NASA Interdisciplinary Consortia for Astrobiology Research (ICAR) Program (80NSSC21K0594).*

*Code availability. The code for the version of the 'muffin' release of the cGENIE Earth system model used in this paper, is tagged as v0.9.13, and is assigned a DOI: 10.5281/zenodo.3999080.*

*Configuration files for the specific experiments presented in the paper can be found in the directory: genie-userconfigs/PUBS/chengetal.BG.2020. Details of the experiments, plus the command line needed to run each one, are given in the readme.txt file in that directory. All other configuration files and boundary conditions are provided as part of the code release.*

*A manual detailing code installation, basic model configuration, tutorials covering various aspects of model configuration, experimental design, and output, plus the processing of results, is assigned a DOI: 10.5281/zenodo.4305997.*

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
