# Peer review of "Characterizing the marine iodine cycle and its relationship to ocean deoxygenation in an Earth System model"

_EGUsphere, 2024_

## Author Comment (AC1)

I come to review this manuscript from a paleo perspective. This study by Cheng et al. has provided fresh and deeper understandings on simulating the marine iodine cycle in the cGENIE model. I find the model ensembles are well designed from both modern and paleo angles. I agree with the authors that the simulations have overall good matches with modern observations and paleo data. The authors have also offered their detailed evaluations on model performance from three perspectives. Their explanations of model-data mismatches are reasonable and have pointed out some future research directions. I think the manuscript is well written and the main points are very clear.

I do have two questions regarding the paleo simulations. It has been speculated that the total iodine concentration in seawater in the geologic past may be different than modern oceans (Zhou et al., 2016 Paleo; Lu et al., 2018 Fig. S12). But the Cretaceous simulations seem to use modern total iodine value? If you used a higher total iodine in the pre-OAE simulations, I assume it will bring all the model-simulated I/Ca to higher values, thus presumably closer to observations?

Thank you for addressing this point. Yes, the total iodine concentration in the Cretaceous cGENIE simulations is 500nM, the same as modern values. Importantly, total iodine can be changed in the model and the impacts of total iodine changes on surface iodate distribution were previously tested in Lu et al., (2018), figure S12 (see below). One interesting outcome is that lower total iodine led to more limited iodate reduction and a relative increase in the proportion of iodate in the surface.

Further, as discussed by Zhou et al., (2015), the total marine iodine budget could have varied due to increased continental weathering or organic carbon burial. However, no such evidence has been reported to date for iodine inventory changes across OAE2.

To clarify this, a brief discussion will be added to L636:

"It is possible that the total iodine inventory has varied through Earth history relative to the modern-day value (~500 nM), which was adopted for our Cretaceous model. Indeed, the overall underestimated I/Ca by cGENIE might be the result of overall higher Cretaceous total iodine inventory (Zhou et al., 2015; Lu et al., 2018). However, such difference is easily masked by local-regional redox variation and is thus difficult to track (Zhou et al., 2015). Due to the lack of evidence otherwise, we assume the average total iodine during the Cretaceous is close to the modern, and the consistent I/Ca

underestimation is caused by uncertainty in model simulation."

[Figure]

**Fig. S12.** Surface-ocean iodate $IO_3^-$ distributions with decreasing ocean iodine inventory. These frequency distributions do not reproduce the lognormal type of distribution as observed in Paleozoic I/Ca data. Modern fluxes of iodine input and output from seawaters may be orders of magnitude lower than that of iodine recycling in the water column (*7*). $IO_3^-$ (instead of $I^-$) sorption on organic matter at the sediment-water interface is a major sink of iodine, which appears to be stabilized by a negative feedback between the amount of organic matter as a substrate on the seafloor and bottom-water oxygenation preventing $IO_3^-$ reduction (*52*). No evidence supports either global organic matter burial rate or bottom water $[O_2]$ level mimicking the I/Ca record. Therefore, secular changes in total iodine concentration are also unlikely to dominate the trends observed in our I/Ca record.

L593-596: The conversion from seawater IO3- and Ca2+ to I/Ca may be more complex than the authors have suggested For example, the substitution of IO3- into calcite may involve Na+, CO3-- ions (Podder et al., 2017 GCA); the seawater Ca2+ concentration in Cretaceous may be different than modern day, so whether Cretaceous Ca2+ is well-simulated needs to be considered. I understand this may be beyond the scope of this model-focused study, but I recommend the authors should at least acknowledge such complications.

Thanks for the very helpful suggestion. An additional comment will be added between L593-596 to address these uncertainties:

"Beyond temperature, we acknowledge that $IO_3^-$ incorporation into carbonate lattice through substitution $IO_3^- + Na^+ \leftrightarrow CO_3^{2-} + Ca^{2+}$ is controlled by $[Na^+]$, $[CO_3^{2-}]$, and $[Ca^{2+}]$ (Podder et al., 2017). However, either quantifying these ions during the Cretaceous seawater or quantitative calculation of ion substitution dynamics requires further constraints. Although uncertainties are inevitable, we assume our temperature- controlled $[IO_3^-]$-I/Ca conversion based on current quantitative knowledge meets the requirement for Cretaceous model-data comparison."

Minor comments:

L65: strictly speaking, it should be "regional rather than in-situ redox conditions"

Will adopt reviewer's suggestion for more accurate wording.

L78: I- re-oxidation

Will correct the typo.

Fig. 6 caption: may add a short note to refer readers to see transect locations shown in Fig. 1

We will either add a note to Fig. 6 caption or change to a new Figure 1 and add a map view to Fig. 6.

L373: strictly speaking, these papers studied both planktic and benthic forams

Will change "benthic foraminiferal" to "planktic and benthic foraminiferal" according to the reviewer's suggestion.

**Citation**: https://doi.org/10.5194/egusphere-2024-677-RC1

---

## Author Comment (AC2)

1. To improve the utility of carbonate I/Ca ratios as a paleotracer requires acknowledgement and quantitative understanding of the fact that dissolved iodine speciation in the ocean is not simply and solely a product of redox conditions. This study addresses this issue by incorporating a range of iodine transformations in the cGENIE model. The representations of iodine cycling are well thought out and appropriate, and cGENIE is a suitable model for paleoreconstructions. The model development and evaluation appear to be well conducted, and the manuscript is well presented. As well as paleo-oceanographers, the iodine model described is also likely to be of interest for those working on present day iodine cycling, such as biogeochemists and air quality modellers.

I have the following major and minor comments on the manuscript:

Major comments:

1. The model is limited in its ability to accurately model present day oxygen levels, and over-estimates levels in the north Pacific OMZ (L317, Fig 6; L513, L537-538). As iodine speciation in the model is a function of oxygen levels, this will affect model performance. Indeed, simulated present day iodine distributions are closer to observations when the model is forced to climatological oxygen values (L340). The accuracy of oxygen predictions for the geological past therefore requires scrutiny. Although the authors briefly allude to this issue (L538-543; L621), a more in depth and up front consideration is required. Is there any way the accuracy of paleo oxygen predictions made by cGENIE be assessed, and the uncertainty associated with this quantified? What future work is planned to reduce the uncertainty in predicted oxygen levels?

> Thanks for the thoughtful question. The accuracy of assessing $O_2$ prediction in the paleo ocean is difficult because of the lack of direct measurements. The goal of this study is to provide a pathway for reconstructing paleo-$O_2$ using I/Ca proxy in order to grasp a full understanding of ocean oxygenation through cross comparison with other geochemical redox proxies.
>
> We acknowledge that the $O_2$ distribution in the modern ETNP (Eastern Tropical North Pacific) between observation and the model has differences in detail. However, the broad $O_2$-depletion feature is simulated correctly. Due to the low special resolution of cGENIE, the whole ETNP only occupies three grids in the model framework. Meanwhile, the depth resolution close to 100-200 m in the surface ocean also limits the finer simulation of ODZ features.
>
> Therefore, cGENIE is better performed in predicting large scale redox distribution in the paleo-ocean (as it is originally designed to) instead of reproducing fine resolution ODZ in the modern.

In order to provide this same input to the readers, we will add the following summary to L543:

In addition, due to the low special resolution of cGENIE, the whole ETNP only occupies three grids in the model framework. Meanwhile, the depth resolution close to 100-200 m in the surface ocean also limits the finer simulation of ODZ features. We are indeed aware of these uncertainties associated with $O_2$ simulation in cGENIE. However, as a model targeted to assist paleoredox reconstructions, broadly reconstructing ODZ features in a coarse spatial resolution is acceptable.

2.  Similarly, to make forward predictions of I/Ca ratios, modelled values of historic calcium concentrations are integral (L594). A brief description of how these are simulated, and discussion of the uncertainty in these values is required.

Thank you for pointing this out. Our current conversion of modeled $[IO_3]$ to I/Ca does not account for [Ca]sw. This is because the impacts of [Ca]sw on the $[IO_3]$-I/Ca relationship have not been quantified. Instead, the 2 calibrations of the relationship only evaluate the impacts of temperature and do not measure or vary [Ca] in the solutions used. Given the current state of knowledge, we used both the temperature and $[IO_3^-]$ from the model to determine I/Ca. Specifically, as described in the supplementary information in Zhou et al., (2014) referenced in line 595, different linear relationships between $[IO_3]$ and I/Ca were observed at 6 °C, 19 °C and 33 °C in a series carbonate synthesis experiment (Figure S3). Then the distribution coefficient $K_D = (I/Ca)/ [IO_3]$ for a given temperature was interpolated based on the linear relationship between temperature and KD (Figure S3, attached below). For our Cretaceous model calibration, we derived the $K_{DS}$ based on local temperatures (at each grid associated with sections) simulated by cGENIE and the linear relationship in Figure S3 in Zhou et al., (2014). We will clarify the details of this transformation in Section 4.3.

[Figure]

[Figure]

3. Figure 7 and L590: State what type of carbonate archive the I/Ca values were measured in. Were these archives likely to have been subject to any diagenetic alterations that could change the I/Ca ratio from that incorporated at the time of calcite synthesis?

Indeed, it is very important to address the post-depositional alteration/diagenesis when discussing I/Ca records. Post-depositional exposure to anoxic pore water could lower I/Ca in carbonates. While contamination from organically bound iodine could result in a false elevated I/Ca. According to Zhou et al., (2015), from which we adopted the I/Ca data, no noticeable iodine contamination was observed in any of the sections. Minor diagenesis was observed in Demerara Rise, Tarfaya, and Raia de Pedale, which were hypoxic sections and primary I/Ca was low. Therefore, the primary I/Ca signals from these sections were not terribly altered. We will add these clarifications into Section 4.3 for a more consistent logical flow.

L592: "Diagenesis of carbonate hosted I/Ca tends to lower the primary values (Hardisty et al., 2017). However, such an offset is hard to quantitatively predict based on our current knowledge. In addition, according to Zhou et al., (2015), from which we adopted the I/Ca data, most of the sections only suffered minor diagenesis. To simplify the Cretaceous I/Ca-$IO_3$ conversion, we regard the measured I/Ca as primary and acknowledge there is potential uncertainty."

Minor comments:

1. At a number of points, the paper states that biologically mediated iodate reduction is assimilatory (L14, Figure 1, L197, L410). However, it is not yet established whether this process is assimilatory or dissimilatory (e.g. Hepach et al., 2020), it may even be a mixture of processes. This should be made clear in the manuscript. Biologically mediated iodate reduction in the model is represented as an assimilatory process, which is reasonable given the current state of knowledge, but it should be made clear that this is an assumption in the model construction.

Additional clarification will be added to line 197.

L197: "Phytoplankton-absorbed iodine is stored in the cell as $IO_3^-$, $I^-$, or other forms, followed by release during senescence (Hepach et al., 2020). While there is some uncertainty as to whether iodate reduction is assimilatory or dissimilatory (Hepach et al., 2020), it is necessary to clarify here  that in order to simplify the simulation, the modeled $IO_3^-$ tracer is assimilated by phytoplankton and incorporated into POM during photosynthesis (Elderfield and Truesdale, 1980) before being released back to the water column as $I^-$ during remineralization (Wong et al., 2002; Hepach et al., 2020; Wong et al., 1985)."

2. In a few places throughout the manuscript (e.g. L77-79, L137-139, L422-424,) minor grammatical and/or wording improvements are needed to make the text more readily understandable.

Words and grammar will be checked and fixed in these lines.

3. L87: This should be "deep" not "dissolved" chlorophyll maximum

Will be changed according to reviewer's suggestion.

4. L139 and elsewhere: Check and correct the spelling of technical terms e.g. 'respiration' and 'saturation'

These terms in L139 should be correct but will check those through the whole manuscript.

5. L149-151: Either add numbers 2-4 to the list of processes here, or remove the "(1)".

Will add numbers to each of the process for easier tracking.

6. L155: It would be helpful to state here that these representations apply to water column reduction (i.e. process 1) and oxidation (i.e. process 2)

Will add the clarifications according to reviewer's suggestion.

7. L209 and SI Table 4: The machine learning model for sea surface iodide concentrations described Sherwen et al., 2019, was built using the data set in Chance et al., 2019, so it is not clear why this paper is referenced here and in SI Table 4? Were simulated iodide values from Sherwen et al., 2019, also used in the model evaluation?

The reviewer is correct that the citation here is incorrect. We will remove the wrong citation of Sherwen et al., (2019). Simulated iodide from Sherwen et al., 2019 was not used in the model.

8. Figure 3 and SI Figure 1: The coloured dots in these figures are very difficult to see, can they be increased in size, and/or the quality of the figures improved?

The symbol size in Figures 3 and S1 will be adjusted larger for better visibility.

9.  L179: The first sentence requires a reference, and/or more explanation, to make clear how this link between iodide and nitrification being considered here differs from the link with ammonium oxidation mentioned on L188. L432-441: Similarly, the reasoning for extending the proposed link between iodide oxidation and bacterial nitrification (L432-436) to a broader possible relationship with bacterial oxidising activity (L441) should be explained in a little more detail.

We agree with the reviewer that a distinction is necessary. To clarify, the link between iodide and nitrite oxidation is hypothetical and has not been explored previously to our knowledge. We will amend the above referenced sentence (see below) to clarify this point in the main text.

L179: "Given the overlapping redox potential between I and N (e.g., Rue et al., 1997; Cutter et al., 2018), we explore the potential for a link between areas of $I^-$ and nitrification. To simulate this, we devise an alternative "Fennel" scheme, in which $I^-$ oxidation rates vary as a function of ambient $O_2$, increasing with ambient $O_2$ concentrations towards some hypothetical maximum value following Michaelis–Menten kinetics (Fennel et al., 2005)."

10. L244: Why were only five different combinations of parameterisations tested, when nine combinations are possible? Please explain why the five tested combinations were selected.

We aim this manuscript to improve iodine cycle simulation based on previous work (Lu et al, 2018) which used lifetime-threshold combination and achieved some agreement with observations (their Figure S5). We tested additional parameterizations based on a combination of observations as well as hypothetical scenarios not yet grounded in field or experimental observation. Importantly, we purposefully chose to only test combinations of iodate reduction and iodide oxidation parameterizations where at least one of the parameterizations is grounded in observation. So far, only "threshold" (Lu et al., 2020) and "lifetime" (Truesdale, 1980) are based on field-based studies. Therefore, it would be reasonable to take the conservative approach through combining one field-based mechanism ("threshold"-reduction or "lifetime"-oxidation) with our novel but hypothetical alternative mechanisms. That said, it is straightforward for future users to combine any combination of parameterizations provided here.

1. L289: Elevated observed iodide concentrations in surface waters at low latitudes are thought to be a function of biologically mediated reduction and strong vertical stratification (allowing the iodide to accumulate). This should be noted in the text, and the ability of the model to account for the impact of vertical mixing on iodine distribution discussed.

   The model does account for vertical mixing processes, which allows for surface water iodide accumulation in the way the reviewer recommends here. This can be seen in a plot of temperature vs latitude, which reveals higher stratification at low latitudes and weaker at high latitudes. This plot will be added into Supplementary Material.

11. Table 1. I think the 'reminO3lifetime- parameters do not need to be given for simulation 2 (as in Table 3).

    Will be corrected based on reviewer's suggestion.

12. Throughout – insert space between numbers and units

    Will be corrected based on reviewer's suggestion.

13. L354: The assessment against I/Ca records has not yet been described at all, so perhaps should not be mentioned here. Consider including it within the main methods and results sections.

    Will add citations (Zhou et al., 2015) to clarify the source of I/Ca data in L354.

14. L360: I think this should be -0.08 not -0.8?

    The typo will be corrected based on reviewer's suggestion.

15. L451: Is this necessarily the case, if iodate reduction in the model is already a function of oxygen concentration?

    Following sentence will be modified to clarify:

L448-451 (original) "This scenario might hence not perform well in replicating the ocean iodine cycling at intervals during the Phanerozoic when ODZs were highly expanded, or particularly during the Precambrian when the majority of the ocean was ferruginous or euxinic and highly depleted in $O_2$ (for example, summarized by Lyons et al., 2014)."

L448-451 (modified) "This scenario might hence not perform well in replicating the ocean iodine cycling at intervals during the Phanerozoic when ODZs were highly expanded, especially when under low $O_2$ while above the $IO_3$ reduction threshold which inhibits $I^-$ oxidation."

16. L478: This sentence implies that temperature is the main driver of primary production, which is misleading – although temperature has some effect on primary production rates, it is not the dominant controlling factor in the surface ocean. The relationship between iodide abundance and temperature reported in Chance et al. 2014, is instead thought to occur due to the relationship between temperature and vertical mixing. This sentence should be rephrased to reflect this more accurately

The sentence will be modified based on reviewer's suggestion for better accuracy.

L477-478 (original): "The pathway of transforming $IO_3^-$ into $I^-$ in these oxidized waters is through primary productivity in the photic zone, which is temperature dependent (Chance et al., 2014)."

L477-478 (modified): "The pathway of transforming $IO_3^-$ into $I^-$ in these oxidized waters is through primary productivity in the photic zone, which resulted in $I^-$ accumulation within the mixed layer (Chance et al., 2014). In the low latitudinal surface ocean, weaker vertical mixing resulting from warmer surface temperature allows $I^-$ accumulation in the shallow mixed layer (Chance et al., 2014)

17. L512: Does "data" here mean model output? Please clarify in the text

The word "data" will be changed to "model output" to clarify.

18. L563: As noted above, I feel that description of the comparison with I/Ca records in this section might be better as part of the main method and results sections, with just the discussion of the findings in section 4.3.

We will introduce the OAE model earlier in the results section. An additional section 3.5 will be added to Results section (since section 2 is model description):

3.5 Ancient iodine: Oceanic Anoxic Event case study

"To assess the ability of cGENIE to reconstruct the iodine cycle in deep-time, we picked the parameters associated with highest M score in each of the ensembles (Table 2) and ran the model under the OAE2 scenario with variable $PO_4$ (Zhou et al., 2015). We calculated the pre-OAE stage modeled I/Ca based on the linear relationship between I:Ca and $IO_3^-$ and temperature from carbonate synthesis experiments (Zhou et al., 2014). Importantly, this relationship is temperature dependent, so the temperature from the relevant model grid point was used for this calculation. The reconstructed I/Ca from each section (Figure 8) was then compared with the simulated I/Ca in the corresponding model grid, and the comparisons are presented by the M score (Figure 7). Note that most of the models reach highest M scores under 0.6-0.8× modern $PO_4$."

19. Figure 8. The caption here appears to incorrectly list more combinations of parameterisations than the three shown.

The additional parameter combinations were included by mistake and will be removed.

20. L623: "DOC remineralisation" as an additional parametrisation variation has not been mentioned in the text before this point, either add an explanation or remove.

This will be removed from the text. The sentence will be modified as:

"Although we tested additional parameters in this study (Table S1), only those combinations listed in Table 1 ("lifetime - threshold", "lifetime-reminO2lifetime", "lifetime - Fennel") perform well in replicating iodine gradients within the modern ocean (Figs. S2-4)."

21. Supplementary Information: A number of figures include "DOC remineralisation" as an additional parametrisation variation, but this is not explained anywhere in the text.

Since "DOC" remineralization neither improves overall model skill score (Table S1) nor replicates any of the modern ocean iodine gradients (Figures S2-S4), we decide not to include this in the text. However, additional explanations of "DOC remineralization" will be added to the caption of Figure S1 where it appears for the first time.

An additional paragraph will be added into Section 4.1 after L363 to briefly demonstrate the role of "DOC remineralization".

"Notably, our M score results also demonstrate a potentially minor role, if any, for iodine cycling with DOC. As described in 2.3.1, the "reminO2lifetime" and the "reminSO4lifetime" scales I- oxidation and $IO_3^-$ reduction with $O_2$ consumption and $SO_4$ reduction during the remineralization of POC (or POC and DOC), respectively. An alternative (with DOC remineralization) scenario was tested to combine the iodine redox reactions with DOC remineralization in cGENIE (Table S1). Compared to the default settings, adding the DOC remineralization-control to the $I^-$ oxidation ("reminO2lifetime") or $IO_3^-$ reduction ("reminSO4lifetime") does not increase the M score of the model. More than the overall M score, the simulation of latitudinal $I^-$ trend in the defaulted no DOC-controlled iodine cycle models performs better in replicating the depth profiles, as well as the $IO_3^-$ depletion feature in the ODZ (discussed in later sections), especially for "reminO2lifetime-threshold" ensembles."

**Citation**: https://doi.org/10.5194/egusphere-2024-677-RC2

---

## Author Response (AR1)

Dear Editors,

Regarding the revisions of manuscript **Characterizing the marine iodine cycle and its relationship to ocean deoxygenation in an Earth System model**, we have agreed with most of reviewers' comments and made associated changes. Importantly, all the reviewer comments regarded addressing clarity. As such, no new model simulations were run and the interpretations are unchanged from the original version. We have, however, made several important changes to address clarity, which we outline below in addition to providing details as part of the response to reviews.

1. As requested, the manuscript has been read and edited many times by all authors to increase writing clarity.
2. As requested, the OAE section is now included in each of the Methods, Results, and Discussion. Formerly, the OAE text was only present in the Discussion, and we agree with reviewers that this made that section overly long and detailed and interrupted the pace of the Discussion. The new Method section 2.5 and Result section 3.5 reflect this change. Most of these changes involved simply moving appropriate text from the Discussion, but some text was added where needed to make the text best formatted for Methods and Results. One outcome is that the Discussion section is now shorter and more straightforward.
3. As requested, more discussion was added regarding the limitations of the oxygen model in cGENIE and which parameterizations are best suited for ancient application. This is now its own new section in the Discussion (Section 4.4).
4. We agreed with the reviewers that the inclusion of some discussion of parameterizations with a DOC remineralization component was confusing. Specifically, these parameterizations didn't perform best, provided no real insights, and were underdeveloped (if at all) so they were only in the supplement. To avoid confusion, we found it most straightforward to just remove these vague references from the text and remove the DOC remineralization-related parameterizations from the manuscript entirely.
5. Figure 1 is now updated to number the processes, which corresponds to numbers given in the text describing these processes.
6. We provide an improved version of Figure 2, which uses a similar format but we think better demonstrates the model gridding process.
7. As requested, Figure 4 (formerly Figure 3) has been updated so that the text and dot sizes are larger and more legible.
8. We added text to Discussion Section 4.3 to address the potential for variable total iodine in the past, as requested by reviewers.
9. We described the method of modeled IO3-I/Ca conversion in Sections 2.5 and 4.3 as well as discussed the associated caveats.

I come to review this manuscript from a paleo perspective. This study by Cheng et al. has provided fresh and deeper understandings on simulating the marine iodine cycle in the cGENIE model. I find the model ensembles are well designed from both modern and paleo angles. I agree with the authors that the simulations have overall good matches with modern observations and paleo data. The authors have also offered their detailed evaluations on model performance from three perspectives. Their explanations of model-data mismatches are reasonable and have pointed out some future research directions. I think the manuscript is well written and the main points are very clear.

I do have two questions regarding the paleo simulations. It has been speculated that the total iodine concentration in seawater in the geologic past may be different than modern oceans (Zhou et al., 2016 Paleo; Lu et al., 2018 Fig. S12). But the Cretaceous simulations seem to use modern total iodine value? If you used a higher total iodine in the pre-OAE simulations, I assume it will bring all the model-simulated I/Ca to higher values, thus presumably closer to observations?

Thank you for addressing this point. Text has been added to section 4.3 L696-700 to address this.

"One possible explanation for the overall underestimation of I/Ca by cGENIE might then be that the Cretaceous iodine inventory was higher than modern (Zhou et al., 2015; Lu et al., 2018). Even a moderate increase (ca. 20-40%) in the ocean iodine inventory (which we did not test here) would presumably act to increase the slope of the regression lines for the parameterization-combinations 'fennel-threshold' and 'reminO2lifetime-threshold' and bring them close to the 1:1 line (Fig. 9)."

L593-596: The conversion from seawater IO3- and Ca2+ to I/Ca may be more complex than the authors have suggested For example, the substitution of IO3- into calcite may involve Na+, CO3-- ions (Podder et al., 2017 GCA); the seawater Ca2+ concentration in Cretaceous may be different than modern day, so whether Cretaceous Ca2+ is well-simulated needs to be considered. I understand this may be beyond the scope of this model-focused study, but I recommend the authors should at least acknowledge such complications.

Thanks for the very helpful suggestion. The associated I/Ca-IO3 conversion has been moved to Methods Section 2.5. An additional comment has been added between L306-310 to address these uncertainties:

"Beyond temperature, we acknowledge that $IO_3^-$ incorporation into carbonate lattice through substitution $IO_3^- + Na^+ \leftrightarrow CO_3^{2-} + Ca^{2+}$ is controlled by $[Na^+]$, $[CO_3^{2-}]$, and $[Ca^{2+}]$ (Podder et al., 2017). However, either quantifying these ions during the Cretaceous seawater or quantitative calculation of ion substitution dynamics requires further

constraints. Although uncertainties are inevitable, we assume our temperature- controlled [IO$_3^-$]-I/Ca conversion based on current quantitative knowledge meets the requirement for Cretaceous model-data comparison."

Minor comments:

L65: strictly speaking, it should be "regional rather than in-situ redox conditions"

Adopted reviewer's suggestion for more accurate wording in L70.

L78: I- re-oxidation

Corrected the typo in L84.

Fig. 6 caption: may add a short note to refer readers to see transect locations shown in Fig. 1

Figure 6 is now figure 7 due to figures rearrangement. We added a note to Fig. 7 caption as well as highlighted the transect in Figure 2.

L373: strictly speaking, these papers studied both planktic and benthic forams

We changed "benthic foraminiferal" to "planktic and benthic foraminiferal" according to the reviewer's suggestion in L464.

**Citation**: https://doi.org/10.5194/egusphere-2024-677-RC1

1. To improve the utility of carbonate I/Ca ratios as a paleotracer requires acknowledgement and quantitative understanding of the fact that dissolved iodine speciation in the ocean is not simply and solely a product of redox conditions. This study addresses this issue by incorporating a range of iodine transformations in the cGENIE model. The representations of iodine cycling are well thought out and appropriate, and cGENIE is a suitable model for paleoreconstructions. The model development and evaluation appear to be well conducted, and the manuscript is well presented. As well as paleo-oceanographers, the iodine model described is also likely to be of interest for those working on present day iodine cycling, such as biogeochemists and air quality modellers.

I have the following major and minor comments on the manuscript:

Major comments:

1. The model is limited in its ability to accurately model present day oxygen levels, and over-estimates levels in the north Pacific OMZ (L317, Fig 6; L513, L537-538). As iodine speciation in the model is a function of oxygen levels, this will affect model performance. Indeed, simulated present day iodine distributions are closer to observations when the model is forced to climatological oxygen values (L340). The accuracy of oxygen predictions for the geological past therefore requires scrutiny. Although the authors briefly allude to this issue (L538-543; L621), a more in depth and up front consideration is required. Is there any way the accuracy of paleo oxygen predictions made by cGENIE be assessed, and the uncertainty associated with this quantified? What future work is planned to reduce the uncertainty in predicted oxygen levels?

   Thanks for the thoughtful question. The accuracy of assessing O2 prediction in the paleo ocean is difficult because of the lack of direct measurements. The goal of this study is to provide a pathway for reconstructing paleo-O2 using I/Ca proxy in order to grasp a full understanding of ocean oxygenation through cross comparison with other geochemical redox proxies.

   To better illustrate this, we have added an additional section 4.4 (Choice of marine iodine cycle representation in cGENIE) to help with the discussion of limitations of applying cGENIE iodine cycle in paleoceanographic research and future directions of model improvement.

2. Similarly, to make forward predictions of I/Ca ratios, modelled values of historic calcium concentrations are integral (L594). A brief description of how these are simulated, and discussion of the uncertainty in these values is required.

Thank you for pointing this out. Our current conversion of modeled [IO3] to I/Ca does not account for [Ca]sw. This is because the impacts of [Ca]sw on the [IO3]-I/Ca relationship have not been quantified. Instead, the 2 calibrations of the relationship only evaluate the impacts of temperature and do not measure or vary [Ca] in the solutions used. Given the current state of knowledge, we used both the temperature and [IO3-] from the model to determine I/Ca. Specifically, as described in the supplementary information in Zhou et al., (2014) referenced in line 595, different linear relationships between [IO$_3$] and I/Ca were observed at 6 °C, 19 °C and 33 °C in a series carbonate synthesis experiment (Figure S3). Then the distribution coefficient $K_D$ = (I/Ca)/ [IO$_3$] for a given temperature was interpolated based on the linear relationship between temperature and KD (Figure S3, attached below). For our Cretaceous model calibration, we derived the $K_{DS}$ based on local temperatures (at each grid associated with sections) simulated by cGENIE and the linear relationship in Figure S3 in Zhou et al., (2014).

We now better describe the modeled [IO3]-I/Ca conversion in section 2.5 (Evaluation against geological observations) and discussed the limitation of this transformation method in Section 4.3 between L703 and L707.

[Figure]

3.  Figure 7 and L590: State what type of carbonate archive the I/Ca values were measured in. Were these archives likely to have been subject to any diagenetic alterations that could change the I/Ca ratio from that incorporated at the time of calcite synthesis?

Indeed, it is very important to address the post-depositional alteration/diagenesis when discussing I/Ca records. Post-depositional exposure to anoxic pore water could lower I/Ca in carbonates. While contamination from organically bound

iodine could result in a false elevated I/Ca. According to Zhou et al., (2015), from which we adopted the I/Ca data, no noticeable iodine contamination was observed in any of the sections. Minor diagenesis was observed in Demerara Rise, Tarfaya, and Raia de Pedale, which were hypoxic sections and primary I/Ca was low. Therefore, the primary I/Ca signals from these sections were not terribly altered. We have added additional discussion of diagenesis in L295.

"Diagenesis of carbonate hosted I/Ca tends to lower the primary values (Hardisty et al., 2017). However, such an offset is hard to quantitatively predict based on our current knowledge. In addition, according to Zhou et al., (2015), from which we adopted the I/Ca data, most of the sections only suffered minor diagenesis. To simplify the Cretaceous I/Ca-to-IO3 conversion, we regard the measured I/Ca as primary and acknowledge there is potential uncertainty."

Minor comments:

1. At a number of points, the paper states that biologically mediated iodate reduction is assimilatory (L14, Figure 1, L197, L410). However, it is not yet established whether this process is assimilatory or dissimilatory (e.g. Hepach et al., 2020), it may even be a mixture of processes. This should be made clear in the manuscript. Biologically mediated iodate reduction in the model is represented as an assimilatory process, which is reasonable given the current state of knowledge, but it should be made clear that this is an assumption in the model construction.

Additional clarification has been added to the Figure 1 caption and line 209.

L209: "Phytoplankton-absorbed iodine is stored in the cell as $IO_3^-$, $I^-$, or other forms, followed by release during senescence (Hepach et al., 2020). While there is some uncertainty as to whether iodate reduction is assimilatory or dissimilatory (Hepach et al., 2020), we adopt a comparable approach to nitrogen cycling (sequence: NO3- uptake, N incorporation into organic matter, remineralization and release as the reduced NH4+ form). We assume that IO3- is assimilated by phytoplankton and incorporated into POM during photosynthesis (Elderfield and Truesdale, 1980) and released as I- during remineralization and/or cell senescence (Wong et al., 2002; Hepach et al., 2020; Wong et al., 1985)."

2. In a few places throughout the manuscript (e.g. L77-79, L137-139, L422-424,) minor grammatical and/or wording improvements are needed to make the text more readily understandable.

   Words and grammar have been checked and fixed throughout the text.

3. L87: This should be "deep" not "dissolved" chlorophyll maximum

   The associated sentence is now in L93. The word has been changed according to reviewer's suggestion.

4. L139 and elsewhere: Check and correct the spelling of technical terms e.g. 'respiration' and 'saturation'

   The term is in L145 now. This should be correct and therefore there is no need to be changed.

5. L149-151: Either add numbers 2-4 to the list of processes here, or remove the "(1)".

   Numbers have been added to each of the process for easier tracking in L152-155.

6. L155: It would be helpful to state here that these representations apply to water column reduction (i.e. process 1) and oxidation (i.e. process 2)

   We have added the clarifications in L159-160 according to reviewer's suggestion.

7. L209 and SI Table 4: The machine learning model for sea surface iodide concentrations described Sherwen et al., 2019, was built using the data set in Chance et al., 2019, so it is not clear why this paper is referenced here and in SI Table 4? Were simulated iodide values from Sherwen et al., 2019, also used in the model evaluation?

The reviewer is correct that the citation here is incorrect. We have removed the wrong citation of Sherwen et al., (2019). Simulated iodide from Sherwen et al., 2019 was not used in the model. (L221)

8. Figure 3 and SI Figure 1: The coloured dots in these figures are very difficult to see, can they be increased in size, and/or the quality of the figures improved?

The symbol sizes in Figures 4 (previously Figure 3) and S1 have been adjusted larger for better visibility.

9. L179: The first sentence requires a reference, and/or more explanation, to make clear how this link between iodide and nitrification being considered here differs from the link with ammonium oxidation mentioned on L188. L432-441: Similarly, the reasoning for extending the proposed link between iodide oxidation and bacterial nitrification (L432-436) to a broader possible relationship with bacterial oxidising activity (L441) should be explained in a little more detail.

We agree with the reviewer that a distinction is necessary. To clarify, the link between iodide and nitrite oxidation is hypothetical and has not been explored previously to our knowledge. We ammended the above referenced sentence (see below) to clarify this point in the main text.

L189: "Given the overlapping redox potential between I and N (e.g., Rue et al., 1997;    Cutter et al., 2018), we explore the potential for a link between areas of I- and nitrification. To simulate this, we devise an alternative "Fennel" scheme, in which I- oxidation rates vary as a function of ambient O2, increasing with ambient

O2 concentrations towards some hypothetical maximum value following Michaelis–Menten kinetics (Fennel et al., 2005)."

10. L244: Why were only five different combinations of parameterisations tested, when nine combinations are possible? Please explain why the five tested combinations were selected.

We aim this manuscript to improve iodine cycle simulation based on previous work (Lu et al, 2018) which used lifetime-threshold combination and achieved some agreement with observations (their Figure S5). We tested additional parameterizations based on a combination of observations as well as hypothetical scenarios not yet grounded in field or experimental observation. Importantly, we purposefully chose to only test combinations of iodate reduction and iodide oxidation parameterizations where at least one of the parameterizations is grounded in observation. So far, only 'threshold' (Lu et al., 2020) and 'lifetime' (Truesdale, 1980) are based on field-based studies. Therefore, it would be reasonable to take the conservative approach through combining one field-based mechanism ('threshold'-reduction or 'lifetime'-oxidation) with our novel but hypothetical alternative mechanisms. That said, it is straightforward for future users to combine any combination of parameterizations provided here.

1. L289: Elevated observed iodide concentrations in surface waters at low latitudes are thought to be a function of biologically mediated reduction and strong vertical stratification (allowing the iodide to accumulate). This should be noted in the text, and the ability of the model to account for the impact of vertical mixing on iodine distribution discussed.

The model does account for vertical mixing processes, which allows for surface water iodide accumulation in the way the reviewer recommends here. An additional Figure S7 (latitudinal distribution of vertical velocity of ocean currents) has been added to help demonstrate water column stratification in low latitudes.

11. Table 1. I think the 'reminO3lifetime- parameters do not need to be given for simulation 2 (as in Table 3).

The typo has been corrected based on reviewer's suggestion.

12. Throughout – insert space between numbers and units

Has been corrected based on reviewer's suggestion.

13. L354: The assessment against I/Ca records has not yet been described at all, so perhaps should not be mentioned here. Consider including it within the main methods and results sections.

We have added the corresponding description of adopting I/Ca from (Zhou et al., 2015) in the new Section 2.5.

14. L360: I think this should be -0.08 not -0.8?

The typo currently in L450 has been corrected.

15. L451: Is this necessarily the case, if iodate reduction in the model is already a function of oxygen concentration?

The sentence is now in L538.

L448-451 (original) "This scenario might hence not perform well in replicating the ocean iodine cycling at intervals during the Phanerozoic when ODZs were highly expanded, or particularly during the Precambrian when the majority of the ocean was ferruginous or euxinic and highly depleted in O2 (for example, summarized by Lyons et al., 2014)."

L538-540 (modified) "This scenario might hence not perform well in replicating the ocean iodine cycling at intervals during the Phanerozoic when ODZs were highly expanded, as it does not account for the possibility for slower I- oxidation at low O2 but above the IO3- reduction O2 threshold."

16. L478: This sentence implies that temperature is the main driver of primary production, which is misleading – although temperature has some effect on primary production rates, it is not the dominant controlling factor in the surface ocean. The relationship between iodide abundance and temperature reported in Chance et al. 2014, is instead thought to occur due to the relationship between temperature and vertical mixing. This sentence should be rephrased to reflect this more accurately

The sentence has been modified based on reviewer's suggestion for better accuracy.

L477-478 (original): "The pathway of transforming IO3- into I- in these oxidized waters is through primary productivity in the photic zone, which is temperature dependent (Chance et al., 2014)."

L567-570 (modified): "The pathway of transforming IO3- into I- in these oxidized waters is through primary productivity in the photic zone, which results in I- accumulation within the mixed layer (Chance et al., 2014). In the low latitudinal surface ocean, weaker vertical mixing resulting from warmer surface temperatures allows I- accumulation in the shallow mixed layer (Chance et al., 2014; Moriyaus et al., 2023)"

17. L512: Does "data" here mean model output? Please clarify in the text

The word "data" has been changed to "model output" to clarify. (Currently in L603)

18. L563: As noted above, I feel that description of the comparison with I/Ca records in this section might be better as part of the main method and results sections, with just the discussion of the findings in section 4.3.

We agree with the reviewer's suggestion. An additional section 2.5 (Evaluation against geological observations) has been added to Model Description.

19. Figure 8. The caption here appears to incorrectly list more combinations of parameterisations than the three shown.

Previous Figure 8 is Figure 9 after rearrangement. The additional parameter combinations have been removed.

20. L623: "DOC remineralisation" as an additional parametrisation variation has not been mentioned in the text before this point, either add an explanation or remove.

We agree that the description of DOC remineralization-related schemes is confusing in manuscript. Specifically, these schemes do not perform well either in overall M-score or replicating the iodine gradients in modern ocean. In addition, there is currently no sufficient evidence that this process participates in I-oxidation in the modern ocean. Due to these reasons, we have agreed to remove all the contents related to 'DOC remineralization' from the text as well as SI.

21. Supplementary Information: A number of figures include "DOC remineralisation" as an additional parametrisation variation, but this is not explained anywhere in the text.

Referring to the explanation in the previous minor comment, we decided to remove the 'DOC remineralization' content.

**Citation**: https://doi.org/10.5194/egusphere-2024-677-RC2